# KEA: Keeping Exploration Alive
# by Proactively Coordinating Exploration Strategies

**Shih-Min Yang** [1]   **Martin Magnusson** [1]   **Johannes A. Stork** [1]   **Todor Stoyanov** [1]

## Abstract

Soft Actor-Critic (SAC) has achieved notable success in continuous control tasks but struggles in sparse reward settings, where infrequent rewards make efficient exploration challenging. While novelty-based exploration methods address this issue by encouraging the agent to explore novel states, they are not trivial to apply to SAC. In particular, managing the interaction between novelty-based exploration and SAC's stochastic policy can lead to inefficient exploration and redundant sample collection. In this paper, we propose KEA (Keeping Exploration Alive) which tackles the inefficiencies in balancing exploration strategies when combining SAC with novelty-based exploration. KEA integrates a *novelty-augmented* SAC with a standard SAC agent, proactively coordinated via a switching mechanism. This coordination allows the agent to maintain stochasticity in high-novelty regions, enhancing exploration efficiency and reducing repeated sample collection. We first analyze this potential issue in a 2D navigation task, and then evaluate KEA on the DeepSea hard-exploration benchmark as well as sparse reward control tasks from the DeepMind Control Suite. Compared to state-of-the-art novelty-based exploration baselines, our experiments show that KEA significantly improves learning efficiency and robustness in sparse reward setups.

## 1. Introduction

Despite the success of deep reinforcement learning (RL) in continuous control tasks, such as robotic manipulation, these methods often rely on manually designed dense rewards (Zhou et al., 2023; Zhou & Held, 2023; Zhang et al., 2023; Yang et al., 2024), which require task-specific expertise and limit task generalization. To reduce reliance on handcrafted dense rewards, early works have focused on sparse reward settings, where feedback is rare. However, this makes exploration difficult, and basic strategies like stochastic sampling (Tokic, 2010; Bridle, 1989) or additive noise (Silver et al., 2014) often fail. In this context, Soft Actor-Critic (SAC) (Haarnoja et al., 2018; Christodoulou, 2019) has shown significant success in continuous control tasks (Zhou et al., 2023; Zhou & Held, 2023; Yang et al., 2022) by optimizing exploration and exploitation via stochastic policies and entropy regularization. However, even SAC struggles with inefficient exploration under sparse rewards.

Reward shaping (Ng et al., 1999; Hu et al., 2020; Ladosz et al., 2022) has been used to improve exploration, but it risks misaligning the agent's behavior from the true task objective (Irpan, 2018; Popov et al., 2017). Solving sparse reward tasks is essential for objective alignment. One promising solution to sparse rewards is to augment (extrinsic) rewards with intrinsic signals that encourage exploration. Curiosity-based methods (Pathak et al., 2017; Burda et al., 2019) use prediction errors from learned dynamics models, while novelty-based methods (Burda et al., 2018; Badia et al., 2020) compute intrinsic rewards based on the novelty of visited states. However, while unvisited states may offer high intrinsic rewards in principle, the agent has no prior experience to estimate their novelty. As a result, novelty-based methods tend to reinforce revisiting previously known states that have already been assigned high intrinsic rewards, lacking a mechanism to actively target truly unvisited states. To improve exploration, NovelD (Zhang et al., 2021) combines novelty differences with episodic counting-based bonuses to focus exploration on the boundary between visited and unvisited regions. This increases the likelihood of entering previously unvisited regions by chance.

While novelty-based methods have been shown to improve exploration when coupled with an on-policy RL method such as PPO (Schulman et al., 2017), applying them to sample-efficient off-policy methods presents additional challenges. For example, Soft Actor-Critic (SAC) drives exploration via stochastic policy sampling with entropy regularization. When combined with novelty-based methods, this can lead to unintended interactions between exploration

[1]Department of AASS, Örebro University, Örebro, Sweden. Correspondence to: Shih-Min Yang <shih-min.yang@oru.se>.

*Proceedings of the 42nd International Conference on Machine Learning*, Vancouver, Canada. PMLR 267, 2025. Copyright 2025 by the author(s).

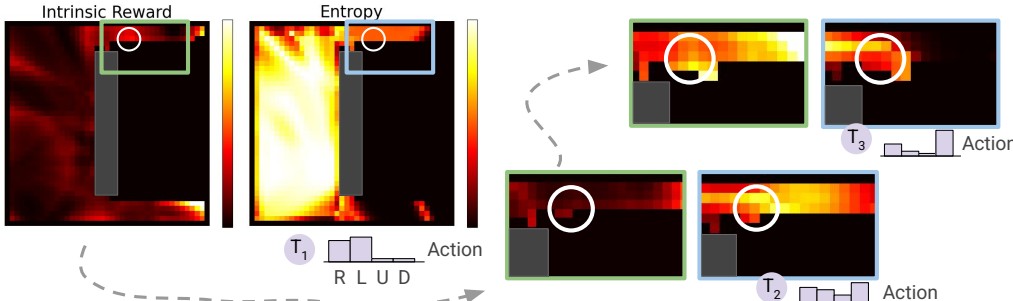

*Figure 1.* **Interactions between exploration strategies**. Each subfigure corresponds to a different training stage and shows the agent's action distribution (*move right*, *left*, *up*, *down*) at a specific region (white circle). The intrinsic reward map visualizes intrinsic rewards (brighter means higher value), and the entropy map reflects policy entropy. The **gray bar** is a fixed obstacle, and the agent's objective is to navigate from the left to the right side of the map. Initially, high intrinsic rewards drive the agent to revisit certain regions repeatedly. As novelty decays, increased policy entropy introduces more randomness, increasing the chance of exploring new areas. This cycle of revisiting and shifting continues, but excessive revisits can lead to redundant experience collection and inefficient learning.

strategies due to the lack of explicit coordination. In particular, novelty-based exploration and SAC's stochastic policy can overlap or interfere, alternately dominating the agent's behavior. Relying solely on natural shifting between them can cause delays in discovering new states: the agent may repeatedly revisit visited states with high intrinsic rewards instead of exploring unvisited ones, leading to redundant experience collection and inefficient exploration (see Appendix A for more details).

To illustrate this dynamic, Fig. 1 shows three representative training stages during training: (1) At time $T_1$, the actions *move right* and *left* have the highest probabilities, and high intrinsic rewards lead the agent to repeatedly revisit a specific region. (2) As intrinsic rewards of this region decay (time $T_2$), policy entropy increases, introducing more randomness into action selection. This shift raises the likelihood of sampling previously unvisited directions, such as *move down*. (3) Once discovering previously unvisited regions (time $T_3$), the agent becomes driven again by high intrinsic rewards to focus on revisiting these newly discovered areas. This cyclical interaction underscores the limitations of relying solely on natural shifting between novelty-based and entropy-driven explorations. Without effective coordination, the agent can collect redundant experiences, slowing down overall learning.

In this paper, we propose KEA (Keeping Exploration Alive) to address inefficiencies arising from the complex interactions between novelty-based exploration and stochastic policy exploration. KEA proactively coordinates different exploration strategies, resulting in consistent exploration behavior, maintaining diversity in exploration, and preventing the agent from repeatedly revisiting explored states. To implement proactive coordination, we integrate a novelty-augmented agent with a standard agent, where the former is guided by novelty-based exploration and the latter preserves stochasticity. A switching mechanism based on state nov-

elty dynamically shifts control between them. Additionally, KEA leverages off-policy RL to collect data using multiple policies. This allows us to use distinct exploration strategies (from the novelty-augmented agent and the standard agent) to gather diverse experiences from the environment.

We evaluate KEA in three experimental settings (Section 3). First, we analyze a 2D navigation task with sparse rewards to study the underlying challenges of novelty-based exploration. Then, we test KEA on the DeepSea hard-exploration benchmark (Osband et al., 2020) and sparse reward control tasks from DeepMind Control Suite (Tassa et al., 2018). In the 2D navigation task, KEA substantially improves learning efficiency by proactively coordinating exploration strategies. Similarly, KEA improves performance over baselines in more challenging tasks. Beyond SAC, we also investigate KEA's generalization to other off-policy methods, including Deep Q-Networks (DQN) (Mnih, 2013) and Soft Q-Learning (SQL) (Haarnoja et al., 2017). Our findings show that KEA is most effective when the original exploration strategy interacts with novelty-based exploration. This demonstrates KEA's potential beyond SAC, highlighting its ability to work across different off-policy RL methods.

Our main contributions are as follows: **(1)** We analyze a potential problem when combining SAC with novelty-based exploration, where the complex interactions between the exploration strategies may cause inefficiencies. **(2)** We propose a method that proactively coordinates exploration strategies, improving exploration efficiency and consistency. KEA is simple to integrate with existing novelty-based exploration methods, offering broad applicability.

## 2. Method

### 2.1. Background

A Markov Decision Process (MDP) is represented by the state $s \in S$, action $a \in A$, transition function $\mathcal{T} : (s, a) \rightarrow$

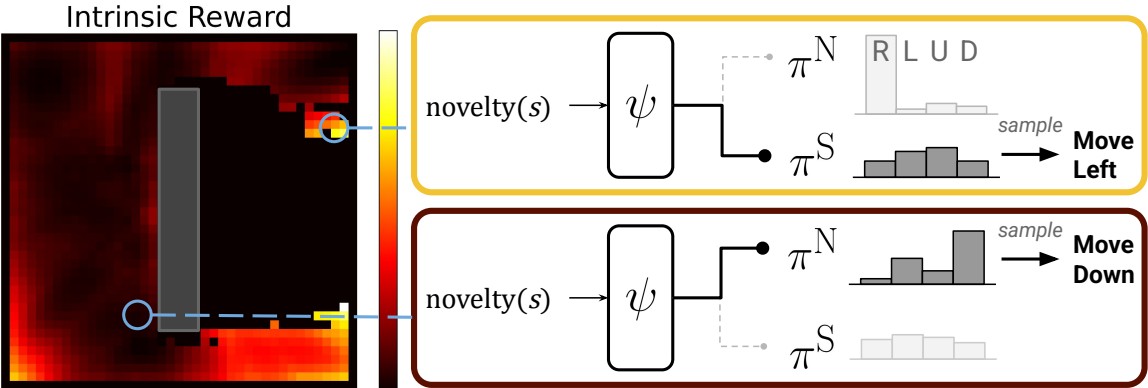

Figure 2. **Overview**. KEA integrates a novelty-augmented SAC ($\mathcal{A}^{\text{N}}$) with a standard SAC agent ($\mathcal{A}^{\text{S}}$). A switching mechanism ($\psi$) proactively coordinates between $\mathcal{A}^{\text{N}}$ and $\mathcal{A}^{\text{S}}$ based on the current state novelty computed by the novelty-based model. The stochastic policies, $\pi^{\text{N}}$ and $\pi^{\text{S}}$, are derived from $\mathcal{A}^{\text{N}}$ and $\mathcal{A}^{\text{S}}$, respectively.

$s'$, reward function $r : S \times A \to \mathbb{R}$, and discount factory $\gamma$. The agent's goal is to find a policy $\pi : S \to A$ that maps the state $s_t$ to the action $a_t$ for maximizing the sum of expected rewards. In this paper, we consider a setup where the primary reward of interest (the "extrinsic" reward) is a sparse binary signal, supplemented by dense "intrinsic" rewards calculated by an intrinsic reward model.

### 2.2. Overview

As Fig. 2, we integrate an additional standard agent (denoted as $\mathcal{A}^{\text{S}}$) with the novelty-augmented agent (denoted as $\mathcal{A}^{\text{N}}$), providing a complementary exploration strategy to address inefficiencies caused by the complexity of interactions between the exploration strategies. To coordinate $\mathcal{A}^{\text{N}}$ and $\mathcal{A}^{\text{S}}$, we devise a switching mechanism, denoted as $\psi$, which dynamically coordinates based on state novelty, measured by the novelty-based model.

In this paper, because SAC has demonstrated significant success in continuous control tasks, we use it as the base RL agent and leverage Random Network Distillation (RND) (Burda et al., 2018) to compute intrinsic reward for exploration ($\mathcal{A}^{\text{N}}$). In an off-policy manner, we can collect transitions with multiple policies while training with another. This allows us to use distinct exploration strategies (e.g. $\mathcal{A}^{\text{N}}$ and $\mathcal{A}^{\text{S}}$) to gather diverse data from the environment. We provide pseudocode for KEA in Algorithm 1.

### 2.3. Exploration Strategies

**Novelty-based Exploration.** Novelty-based exploration encourages the agent to focus on novel states within the explored region, increasing the chances of discovering previously unvisited areas. In this paper, we use SAC as the base RL agent and leverage RND to compute intrinsic rewards that guide this exploration (denoted as $\mathcal{A}^{\text{N}}$). Specifically, the SAC policy is updated to account for both extrinsic rewards

---

**Algorithm 1** KEA: Keeping Exploration Alive

1: Initialize $r^{\text{int}} \leftarrow 0$, replay buffer $\mathcal{D}$, agents $\mathcal{A}^{\text{S}}, \mathcal{A}^{\text{N}}$, and intrinsic reward (IR) model
2: **for** $i = 1$ to (# total steps) **do**
3:     Select policy via switching function $\psi(r^{\text{int}}, \pi^{\text{N}}, \pi^{\text{S}})$
4:     Sample action from selected policy $\pi(\cdot)$ and collect transition from the environment
5:     Compute intrinsic reward $r^{\text{int}}$ via IR model
6:     Store transition in buffer $\mathcal{D}$
7:     Update IR model with latest transitions
8:     **if** $i \in train\_steps$ **then**
9:         Sample batch transitions from $\mathcal{D}$
10:       Recompute intrinsic rewards
11:       Update $\mathcal{A}^{\text{S}}$ and $\mathcal{A}^{\text{N}}$ using SAC
12:     **end if**
13: **end for**

---

(from the environment) and intrinsic rewards (based on novelty). We modify the Soft Bellman update target for the Q network in SAC (Haarnoja et al., 2018) as shown below:

$$y_Q = (\beta^{\text{ext}} \, r^{\text{ext}} + \beta^{\text{int}} \, r^{\text{int}}) + \gamma \left( \min_{\theta'_{1,2}} Q_{\theta'_i}(s', a') - \alpha \log \pi^{\text{SAC}}(\cdot | s') \right) \quad (1)$$

where $\beta^{\text{ext}}$ and $\beta^{\text{int}}$ are scaling factors for extrinsic and intrinsic rewards, and $\alpha$ is the temperature parameter controlling the entropy regularization. The $r^{\text{int}}$ is an intrinsic reward computed based on the state novelty, which measures the prediction error of Random Network Distillation (RND), calculated as

$$r_t^{\text{int}} = ||\hat{f}(s_t; \theta) - f(s_t)||^2, \quad (2)$$

where $f : \mathcal{O} \to \mathbb{R}^K$ represents a randomly initialized target network that maps an observation $s_t$ to an embedding in

$\mathbb{R}^K$, and $\hat{f} : \mathcal{O} \to \mathbb{R}^K$ is a predictor network trained via gradient descent to minimize the expected mean squared error (MSE) with the target network.

**Stochastic Policy via Standard Agent.** We integrate an additional standard agent (denoted as $\mathcal{A}^S$) alongside the novelty-augmented agent ($\mathcal{A}^N$), offering a complementary exploration strategy characterized by high stochasticity. Specifically, $\mathcal{A}^S$ includes a stochastic policy that maintains high action variance by delaying learning until extrinsic rewards are obtained. This is implemented by assigning zero weights to its losses, effectively freezing gradient updates. Once extrinsic rewards are received, the full losses are applied, allowing the agent to begin learning from accumulated experiences in the replay buffer.

With $\mathcal{A}^S$ maintaining high variance in its actions, we proactively coordinate $\mathcal{A}^N$ and $\mathcal{A}^S$ to prevent reliance solely on natural shifts in exploration strategies caused by changes in entropy and intrinsic rewards. This coordination ensures consistent exploration of unvisited regions by reducing deterministic actions in regions of high novelty but low entropy.

In this paper, we implement $\mathcal{A}^S$ using another SAC agent to enhance data efficiency by sharing experiences in a unified replay buffer with $\mathcal{A}^N$. During training, experiences are sampled from this shared buffer, and the policy and Q-networks of $\mathcal{A}^S$ are updated concurrently with those of $\mathcal{A}^N$. The notable difference is that the standard agent is trained using a different reward signal, only taking into account the primary (sparse reward) task, which allows it to retain high entropy for the exploration of unvisited states.

### 2.4. Switching Mechanism

Since our method involves two exploration strategies from different agents, we require a mechanism to determine when to use each. The role of the switching mechanism is crucial for proactively coordinating $\mathcal{A}^N$ and $\mathcal{A}^S$. Simply averaging the action distributions from both agent policies would not be effective, as their objectives may differ significantly. Instead, we design a switching mechanism that adapts based on the novelty of the agent's current state. This mechanism ensures that $\mathcal{A}^S$ operates near the boundary between explored and unexplored regions, while $\mathcal{A}^N$ frequently revisits relatively novel states within the explored regions.

We define the switching criterion as follows:

$$\pi(s_t) = \psi(r_t^{\text{int}}, \pi^N(s_t), \pi^S(s_t)), \qquad (3)$$

$$\psi = \begin{cases} \pi^S(s_t) & \text{, if } r_t^{\text{int}} > \sigma \\ \pi^N(s_t) & \text{, otherwise} \end{cases} \qquad (4)$$

where $\pi^S$ and $\pi^N$ are stochastic policies from $\mathcal{A}^S$ and $\mathcal{A}^N$, respectively, and $\sigma$ is a threshold hyperparameter. When the received intrinsic reward falls below the predefined threshold, the agent switches to $\mathcal{A}^N$ for novelty-based exploration,

which encourages the agent to visit relatively novel areas more often. Conversely, when the received intrinsic reward exceeds the threshold, the agent switches to $\mathcal{A}^S$, focusing on stochastic policy exploration to enter unexplored regions. This switching mechanism provides a proactive coordination of exploration strategies, further improving the exploration efficiency.

## 3. Experiments

In this section, we evaluate the KEA's performance in several RL tasks with sparse rewards to demonstrate its ability to manage the complex interactions between different exploration strategies and improve overall exploration efficiency.

We begin by testing our method on a 2D Navigation task, where the agent must navigate to a fixed goal position while avoiding obstacles. Next, we analyze the sensitivity of KEA to the switching threshold ($\sigma$), a critical hyperparameter that affects the balance between $\mathcal{A}^S$ and $\mathcal{A}^N$. To evaluate KEA's generalization to other off-policy RL methods, we conduct additional experiments by applying KEA to methods such as Deep Q-Networks (DQN) (Mnih, 2013) and Soft Q-Learning (SQL) (Haarnoja et al., 2017). Finally, we evaluate KEA on more challenging environments from the DeepSea hard-exploration benchmark (Osband et al., 2020) and sparse reward control tasks from DeepMind Control Suite (Tassa et al., 2018).

We use Soft Actor-Critic (SAC) as the base RL agent and demonstrate the flexibility of our method by integrating it with two different novelty-based exploration methods. Specifically, we combine SAC with **Random Network Distillation (RND)** (Burda et al., 2018), denoted as **KEA-RND-SAC**, and also with **NovelD** (Zhang et al., 2021), denoted as **KEA-NovelD-SAC**. Each method is evaluated across five random seeds, with results presented as the mean and standard deviation of episodic return. The primary evaluation metric is mean episodic return, which reflects both task performance and convergence speed.

### 3.1. 2D Navigation Task

**Task Description.** As shown in Fig. 3, the 2D Navigation task involves navigating an agent to a fixed goal position on the right (blue point) while avoiding an obstacle placed in the middle of the environment. The agent's starting position (green point) is randomly initialized within the left half of the environment at the beginning of each episode. The environment provides sparse extrinsic rewards, meaning the agent only receives extrinsic rewards when it successfully reaches the goal.

The observation space is discrete, consisting of the agent's current $(x, y)$ position, while the action space includes four possible actions: *(move right, move left, move up, move*

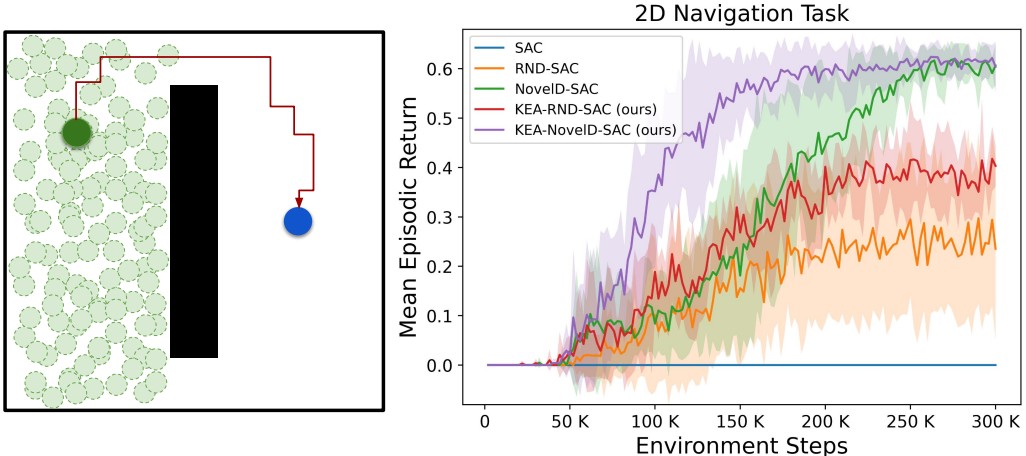

*Figure 3. Left*: 2D Navigation task involves navigating an agent from a randomly chosen start (light green circles) to a fixed goal position on the right (blue point) while avoiding an obstacle placed in the middle of the environment. *Right*: Mean episodic returns during training. The shaded area spans one standard deviation.

| Method | Episodic Return |
|---|---|
| SAC | $0. \pm 0.$ |
| RND-SAC | $0.235 \pm 0.184$ |
| KEA-RND-SAC (ours) | **$0.403 \pm 0.042$** |
| NovelD-SAC | **$0.607 \pm 0.042$** |
| KEA-NovelD-SAC (ours) | $0.604 \pm 0.051$ |

*Table 1.* Mean episodic return (mean±std.) in 2D Navigation task.

| Threshold | Mean Episodic Return | $\mathcal{A}^S$ Usage |
|---|---|---|
| 0.50 | $0.358 \pm 0.151$ | 0.24 |
| 0.75 | $0.348 \pm 0.033$ | 0.19 |
| 1.00 | **$0.407 \pm 0.055$** | 0.14 |
| 1.25 | $0.348 \pm 0.150$ | 0.11 |
| 1.50 | $0.334 \pm 0.167$ | 0.08 |

*Table 2.* Mean episodic return (mean±std.) of KEA-RND-SAC with different switching thresholds.

*down).* Additionally, the transition function is unknown, and the agent must learn to navigate the environment through trial and error. We implement this environment by Gymnasium (Towers et al., 2024).

**Experimental Setup.** In this experiment, we compare the performance of our method (KEA-RND-SAC and KEA-NovelD-SAC) against standard SAC, as well as SAC augmented with novelty-based exploration using RND (RND-SAC) and NovelD (NovelD-SAC). Performance is measured by the mean episodic return during training. The training is halted after the agent collects 300,000 transitions from the environment. Our method variants — KEA-RND-SAC and KEA-NovelD-SAC — use RND and NovelD, respectively, to compute the intrinsic rewards, combined with $\mathcal{A}^S$ and a dynamic switching mechanism to coordinate exploration strategies. Each method is tested across five random seeds, and we report both the mean and standard deviation of the performance to ensure statistical significance.

**Experimental Results.** As shown in Fig. 3, our method significantly outperforms the baselines. The final performance metrics are summarized in Table 1. KEA-RND-SAC achieves a mean episodic return of $0.403 \pm 0.042$ after 300,000 environment steps, compared to RND-SAC's $0.235 \pm 0.184$, representing a more than 70% improve-

ment in performance. In the NovelD setup, NovelD-SAC reaches a mean episodic return of $0.607 \pm 0.042$, while KEA-NovelD-SAC achieves $0.604 \pm 0.051$ after 300,000 environment steps. Although the final performance between KEA-NovelD-SAC and NovelD-SAC is similar, KEA-NovelD-SAC converges significantly faster, reaching a return of 0.6 around 190,000 environment steps, whereas NovelD-SAC requires 250,000 steps to achieve a similar return. These results suggest that KEA's more efficient exploration leads to faster learning efficiency.

### 3.2. Different Switching Thresholds

The switching threshold ($\sigma$) is a critical hyperparameter in KEA that determines the usage of $\mathcal{A}^S$ during training, influencing the balance between $\mathcal{A}^N$ and $\mathcal{A}^S$.

To analyze KEA's sensitivity to different switching thresholds ($\sigma$), we evaluate KEA-RND-SAC's mean episodic return on the 2D Navigation task (Section 3.1). As shown in Table 2, varying the switching threshold affects KEA's performance. Notably, the highest mean episodic return is achieved at $\sigma = 1.0$, with a value of $0.407 \pm 0.055$. While performance slightly drops at other thresholds, all tested configurations consistently outperform the baseline RND-SAC ($0.235 \pm 0.184$). This demonstrates that KEA

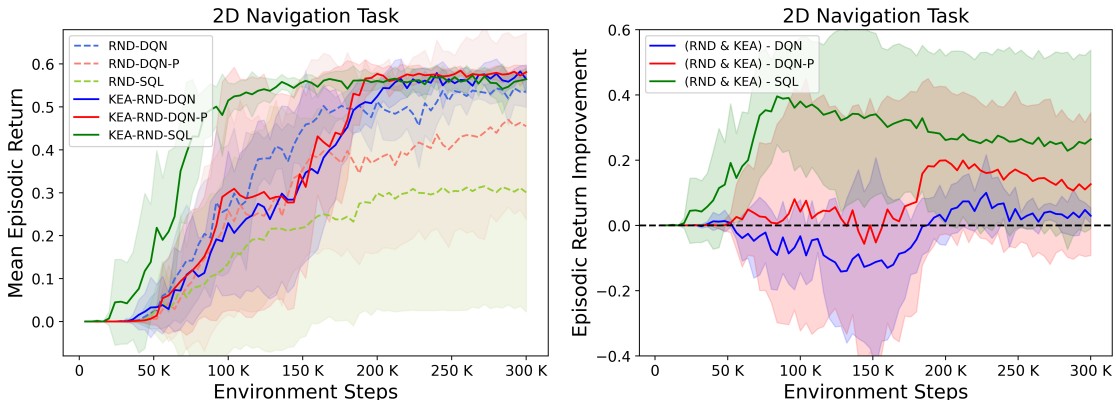

*Figure 4.* **Performance comparison of KEA across off-policy RL methods**. *Left*: Mean episodic returns during training (shaded area represents one standard deviation). *Right*: Improvement in episodic return from KEA across various off-policy RL methods. **(RND & KEA) - DQN** represents the gain from applying KEA to DQN (KEA-RND-DQN minus RND-DQN). Similarly, **(RND & KEA) - DQN-P** and **(RND & KEA) - SQL** denote improvements in DQN-P and SQL. KEA improves exploration in RND-SQL and RND-DQN-P, where the original exploration strategy interacts with novelty-based exploration, but has minimal impact in RND-DQN due to independent $\epsilon$-greedy exploration.

maintains robust performance across a range of threshold values.

To better understand KEA's switching behavior, we analyze $\mathcal{A}^S$ usages across different thresholds (Table 2, column $\mathcal{A}^S$ Usage). As $\sigma$ increases, the usage of $\mathcal{A}^S$ steadily decreases, dropping from 0.24 at $\sigma = 0.50$ to 0.08 at $\sigma = 1.50$. This indicates that a higher threshold restricts $\mathcal{A}^S$ to states with very high intrinsic rewards, resulting in more frequent reliance on $\mathcal{A}^N$. These findings highlight the trade-off between exploration strategies influenced by $\sigma$. Lower thresholds encourage greater usage of $\mathcal{A}^S$, promoting more stochastic exploration, while higher thresholds focus on $\mathcal{A}^N$, leading to more reliance on novelty-based exploration.

### 3.3. Generalization to Other Off-Policy RL Methods

In this paper, we primarily focus on SAC, as the interaction between SAC's stochastic exploration and novelty-based exploration can lead to inefficiencies, which KEA addresses, as demonstrated in Section 3.1. However, a natural question arises: can KEA generalize to other off-policy RL methods? To investigate this, we conducted additional experiments applying KEA to two off-policy methods, Deep Q-Networks (DQN) (Mnih, 2013) and Soft Q-Learning (SQL) (Haarnoja et al., 2017).

DQN employs $\epsilon$-greedy exploration, where actions are selected randomly with probability $\epsilon$, independent of the current best Q-value. In contrast, SQL uses stochastic sampling, where action selection is influenced by all Q-values in the state. To facilitate direct comparison, we further modified DQN's $\epsilon$-greedy exploration to use proportional sampling, where actions are sampled based on their Q-values rather than uniformly. With this modification, the $\epsilon$-greedy explo-

ration becomes dependent on Q-values in the state.

**Experimental Setup.** We evaluate six configurations: (1) RND-DQN and RND-SQL, incorporating Random Network Distillation (RND). (2) KEA-RND-DQN and KEA-RND-SQL, applying KEA to RND-DQN and RND-SQL. (3) RND-DQN-P and KEA-RND-DQN-P, modifying $\epsilon$-greedy exploration with proportional sampling. As in previous experiments, training is halted after the agent collects 300,000 transitions, and each method is tested across five random seeds.

**Experimental Results.** As shown in Fig. 4, our results provide key insights into KEA's generalization across different off-policy RL methods. In the RND-DQN baseline, KEA (KEA-RND-DQN) does not improve performance because $\epsilon$-greedy selects actions with a fixed probability, independent of novelty. As a result, the two exploration strategies do not interact, and KEA's switching mechanism has a limited effect. Since KEA is designed to coordinate exploration strategies that influence each other, its impact is minimal in this setting.

When proportional sampling is introduced in $\epsilon$-greedy (RND-DQN-P), performance drops due to the emerging interaction between exploration strategies. However, applying KEA in this setting (KEA-RND-DQN-P) improves performance by proactively coordinating proportional sampling and novelty-based exploration.

In SQL-RND, KEA (KEA-RND-SQL) significantly improves exploration efficiency. SQL, like SAC, maintains a stochastic policy that interacts with novelty-based exploration, leading to balance shifts similar to those observed in SAC. By proactively coordinating these strategies, KEA

| Algorithm | DeepSea 10 | DeepSea 14 | DeepSea 20 | DeepSea 24 | DeepSea 30 |
|---|---|---|---|---|---|
| DeRL-A2C (Schäfer et al., 2021) | $0.98 \pm 0.10$ | $0.65 \pm 0.23$ | $0.42 \pm 0.16$ | $0.07 \pm 0.10$ | $0.09 \pm 0.08$ |
| DeRL-PPO (Schäfer et al., 2021) | $0.61 \pm 0.20$ | $0.92 \pm 0.18$ | $-0.01 \pm 0.01$ | $0.63 \pm 0.27$ | $-0.01 \pm 0.01$ |
| DeRL-DQN (Schäfer et al., 2021) | $0.98 \pm 0.09$ | $0.95 \pm 0.17$ | $0.40 \pm 0.08$ | $0.53 \pm 0.27$ | $0.10 \pm 0.10$ |
| SOFE-A2C (Castanyer et al., 2024) | $0.94 \pm 0.19$ | $0.45 \pm 0.31$ | $0.11 \pm 0.25$ | $0.08 \pm 0.14$ | $0.04 \pm 0.09$ |
| SOFE-PPO (Castanyer et al., 2024) | $0.77 \pm 0.29$ | $0.67 \pm 0.33$ | $0.13 \pm 0.09$ | $0.07 \pm 0.15$ | $0.09 \pm 0.23$ |
| SOFE-DQN (Castanyer et al., 2024) | $0.97 \pm 0.29$ | $0.78 \pm 0.21$ | $0.70 \pm 0.28$ | $0.65 \pm 0.26$ | $0.42 \pm 0.33$ |
| SAC | $0.98 \pm 0.01$ | $0.69 \pm 0.23$ | $0.00 \pm 0.00$ | $0.00 \pm 0.00$ | $0.00 \pm 0.00$ |
| RND-SAC | $\mathbf{0.99 \pm 0.01}$ | $0.92 \pm 0.13$ | $0.89 \pm 0.09$ | $0.67 \pm 0.35$ | $0.35 \pm 0.44$ |
| **KEA-RND-SAC** (ours) | $\mathbf{0.99 \pm 0.01}$ | $\mathbf{0.99 \pm 0.01}$ | $\mathbf{0.92 \pm 0.05}$ | $\mathbf{0.81 \pm 0.18}$ | $\mathbf{0.54 \pm 0.32}$ |

*Table 3.* Average performance on DeepSea environments of varying sizes, reported with one standard deviation over 100,000 training episodes.

enhances exploration consistency, resulting in better performance.

These findings suggest that KEA is most effective in scenarios where the base exploration strategy is coupled with novelty-based exploration. Specifically, KEA demonstrates the following order of effectiveness: (1) entropy-regularized (probabilistic) policies, such as SAC and SQL, (2) Q-value proportional exploration within a non-greedy exploration step, as in the modified DQN-P, and (3) basic $\epsilon$-greedy exploration, where its impact is minimal. This hierarchy reflects the degree of interaction between the original exploration strategy and novelty-based exploration, which KEA is designed to coordinate.

### 3.4. DeepSea: A Hard-Exploration Benchmark

**Task Description.** The DeepSea environment (Osband et al., 2020) is a well-established benchmark for evaluating hard-exploration challenges in reinforcement learning. It consists of an $N \times N$ grid where the agent starts in the top-left corner and aims to reach a goal in the bottom-right cell. At each timestep, the agent moves one row down and selects an action to shift either left or right. The observation space is a one-hot encoding of the agent's location in the grid, while the action space is discrete (*go left* or *go right*).

The reward function is intentionally deceptive: moving left yields zero reward, while moving right receives a small negative reward of $-0.01/N$. However, the agent receives a sparse reward of $+1$ only when reaching the goal. Each episodes last exactly $N$ steps. Increasing $N$ further increases the difficulty, making DeepSea suitable for evaluating the efficiency of exploration strategies.

**Experimental Setup.** In this experiment, we follow the setup in DeRL (Schäfer et al., 2021) and SOFE (Castanyer et al., 2024) to evaluate KEA and baseline methods on DeepSea environments of increasing size ($N = 10, 14, 20, 24, 30$) to study performance under varying ex-

ploration complexity. Agents are trained for 100,000 episodes, with evaluation every 1000 episodes. We compute the average return across 100 test episodes, providing nsight into both achieved performance and sample efficiency.

All methods are tested across five random seeds per environment size, and we report the mean and standard deviation of the performance to ensure statistical significance. The baselines include standard SAC, SAC with RND (RND-SAC), DeRL, and SOFE.

**Experimental Results.** Table 3 summarizes the average performance of all evaluated methods across DeepSea environments of increasing size ($N = 10$ to $30$). As the environment depth increases, the exploration becomes more challenging due to longer episode lengths and deceptive short-term rewards.

Overall, **KEA-RND-SAC**, consistently achieves competitive or superior performance across all settings, with particularly strong results in the more difficult configurations. Compared to SOFE and DeRL variants, KEA-RND-SAC demonstrates improved sample efficiency and robustness in deeper environments. These findings support the effectiveness of KEA's proactive coordination mechanism, which balances novelty-based and stochastic exploration to enhance performance in sparse-reward settings.

### 3.5. DeepMind Control Suite

**Task Description.** The DeepMind Control Suite (Tassa et al., 2018) is a set of continuous control tasks to evaluate RL algorithms. These tasks simulate various physical environments and require agents to learn complex motor skills to achieve specified goals. Observation spaces are continuous, consisting of joint positions and velocities, while action spaces are represented as continuous values (e.g., joint torques or forces). The number of observation and action dimensions depends on the specific task.

**Experimental Setup.** In this experiment, we compare the

| Method | Walker Run Sparse | Cheetah Run Sparse | Reacher Hard Sparse |
|---|---|---|---|
| SAC | $0. \pm 0.$ | $0. \pm 0.$ | $715.17 \pm 216.57$ |
| RND-SAC | $287.65 \pm 334.12$ | $512.02 \pm 466.26$ | $790.32 \pm 143.26$ |
| KEA-RND-SAC (ours) | $\mathbf{629.74 \pm 196.75}$ | $\mathbf{773.76 \pm 162.74}$ | $\mathbf{874.61 \pm 94.58}$ |
| NovelD-SAC | $553.26 \pm 191.03$ | $647.29 \pm 382.58$ | $\mathbf{860.40 \pm 76.15}$ |
| KEA-NovelD-SAC (ours) | $\mathbf{706.47 \pm 389.23}$ | $\mathbf{734.67 \pm 316.95}$ | $837.12 \pm 68.95$ |

*Table 4.* Mean episodic return (mean±std.) in three tasks from the DeepMind Control Suite.

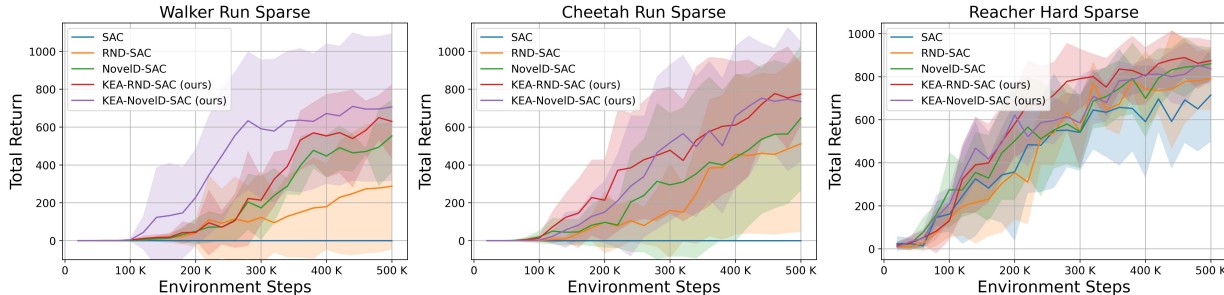

*Figure 5.* Performance on three continuous control tasks from the DeepMind Control Suite. Our method (KEA-RND-SAC and KEA-NovelD-SAC) performs notably better than baselines in more challenging exploration tasks. The shaded regions indicate one standard deviation across evaluation runs.

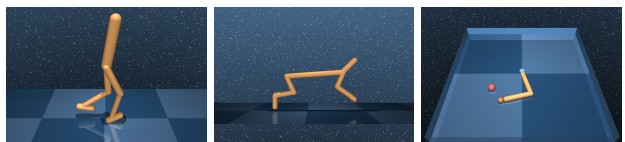

*Figure 6.* Three tasks from the DeepMind Control Suite (Tassa et al., 2018) are used for evaluation: **Walker Run**, **Cheetah Run**, and **Reacher Hard**. The objective in the first two tasks is to run as fast as possible, while in the third task, the agent must reach a specified goal position.

performance of our method (KEA-RND-SAC and KEA-NovelD-SAC) against three baselines: standard SAC, SAC with RND (RND-SAC), and SAC with NovelD (NovelD-SAC). The training is halted after the agent collects 500,000 transitions from the environment. As described earlier in 3.1, KEA-RND-SAC and KEA-NovelD-SAC incorporate an additional standard agent ($\mathcal{A}^S$) and a dynamic switching mechanism to proactively coordinate exploration strategies. They use RND and NovelD, respectively, to compute intrinsic rewards. Each method is tested across five random seeds, and we report both the mean and standard deviation of the performance to ensure statistical significance.

We evaluate the methods on three tasks from the DeepMind Control Suite: **Walker Run Sparse**, **Cheetah Run Sparse**, and **Reacher Hard Sparse** (shown in Fig. 6). In Reacher Hard Sparse, the reward structure is originally sparse. For Walker Run Sparse and Cheetah Run Sparse, rewards are provided sparsely only when the original reward exceeds a certain threshold. The threshold for Walker Run is set at 0.3,

while for Cheetah Run, it is 0.35.

**Experimental Results.** As shown in Fig. 5, Walker Run Sparse and Cheetah Run Sparse present significant exploration challenges. Without novelty-based exploration, SAC struggles to reach the goal. In contrast, Reacher Hard Sparse is relatively easier, as SAC can reach the goal even without intrinsic rewards. Besides, novelty-based exploration improves performance across all three tasks, and our method further enhances this performance.

As shown in Table 4, after 500,000 environment steps, KEA-RND-SAC achieves significant improvements over RND-SAC, with increases of $119\%$, $51\%$, and $11\%$ in mean episodic returns across the three tasks. Similarly, KEA-NovelD demonstrates approximately a $10\%$ improvement over NovelD. Although KEA-NovelD shows similar results to NovelD on the Reacher Hard Sparse task, it performs notably better in the more challenging exploration tasks, Walker Run Sparse and Cheetah Run Sparse.

## 4. Related Work

Computing novelty to improve exploration has emerged as a critical component for improving exploration efficiency in sparse reward setups, where extrinsic rewards are limited (Ladosz et al., 2022; Burda et al., 2019; Kim et al., 2019). These methods complement our work, as KEA can integrate with various curiosity- and novelty-based explorations.

**Prediction Error-based Novelty.** One popular approach is

prediction error-based novelty, which measures state novelty by predicting the next state and calculating the error. Stadie et al.(Stadie et al., 2015) compute the error between the predicted and the actual state in the latent space, while ICM (Pathak et al., 2017) measures the prediction error of an agent's ability to anticipate action outcomes in a learned feature space using a self-supervised inverse dynamics model. RND (Burda et al., 2018) computes state novelty using the prediction error of a randomly initialized network.

**Count-based Novelty.** Count-based novelty methods offer another effective strategy by measuring the novelty based on state visitation frequency. Early works (Bellemare et al., 2016; Ostrovski et al., 2017; Tang et al., 2017) use pseudo-counts to estimate state visitation in high-dimensional environments. Machado et al. (Machado et al., 2020) improves upon earlier methods by using the norm of the successor representation for implicit state counts without requiring domain-specific density models.

**Including Episodic Memory.** Some approaches combine episodic memory and lifelong novelty. For example, NGU (Badia et al., 2020) encourages exploration across episodes and throughout training. RIDE (Raileanu & Rocktäschel, 2020) combines forward and inverse dynamics models with episodic count-based novelty to compute intrinsic rewards based on the distance between consecutive observations in the state embedding space. AGAC (Flet-Berliac et al., 2021) integrates episodic count-based novelty with the KL-divergence between the agent's policy and an adversarial policy to compute intrinsic rewards.

NovelD (Zhang et al., 2021) combines count-based novelty and novelty difference to encourage uniform and boundary exploration, showing strong results in sparse reward tasks. In this paper, we propose KEA, leveraging NovelD for intrinsic rewards while introducing an additional standard agent and a switching mechanism to proactively coordinate exploration strategies and improve efficiency.

**Other Exploration Methods.** Beyond prediction and count-based novelty approaches, other exploration methods include adding noise to parameters (Fortunato et al., 2018; Plappert et al., 2017), computing intrinsic rewards via hierarchical RL (Kulkarni et al., 2016), using curriculum learning to guide exploration (Bengio et al., 2009; Portelas et al., 2021), combining self-supervised reward-shaping methods and count-based intrinsic reward (Devidze et al., 2022), using distance-based metrics for reward shaping (Trott et al., 2019), diversifying policies by regularizing the loss function with distance metrics (Hong et al., 2018), and combining a novelty-based exploration method with switching controls to determine which states to add shaping rewards in a multi-agent RL framework (Zheng et al., 2021).

## 5. Conclusion

In this paper, we present **KEA**, a novel approach to address the challenges of applying novelty-based exploration in SAC. KEA introduces a standard agent ($\mathcal{A}^{\text{S}}$) alongside a novelty-augmented SAC ($\mathcal{A}^{\text{N}}$), which incorporates existing methods such as RND and NovelD. A dynamic switching mechanism proactively coordinates the two exploration strategies, enabling consistent discovery of new regions while maintaining an effective balance between them. Compared to previous methods that rely solely on intrinsic rewards, KEA reduces the complexity arising from the interactions between the novelty-based exploration strategy and the stochastic policy exploration strategy, leading to a more stable training process. Our experiments on DeepSea highlight the effectiveness of KEA's proactive coordination mechanism in balancing novelty-based and stochastic exploration under sparse rewards. Results on sparse reward tasks from the DeepMind Control Suite further demonstrate KEA's substantial improvement over RND-SAC and NovelD-SAC, underscoring its effectiveness in balancing different exploration strategies.

While KEA offers several advantages, one limitation is that it is restricted to off-policy learning, as the additional standard agent shares experiences with the target policy. Nevertheless, we believe KEA provides a principled approach to balancing exploration strategies, advancing exploration in complex environments.

## Acknowledgements

This work received funding from the European Union's Horizon 2020 research and innovation programme under grant agreement No 101017274 (DARKO), and was supported by the Wallenberg AI, Autonomous Systems and Software Program (WASP) funded by the Knut and Alice Wallenberg Foundation. We gratefully acknowledge Yufei Zhu and Finn Rietz for their valuable feedback and insightful discussions.

## Impact Statement

This paper presents work whose goal is to solve the potential issue when combining Soft Actor-Critic and novelty-based exploration methods in the sparse reward environment, advancing the field of Reinforcement Learning. There are many potential societal consequences of our work, none of which we feel must be specifically highlighted here.

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

# A. Theoretical Example: Inefficiency Arising from Interaction Between Exploration Strategies

**Problem Setup and Assumptions.** Consider an MDP with states $S = \{s_0, s_1, s_2\}$ and actions $A = \{a_1, a_2\}$, with deterministic transitions: $T(s_1|s_0, a_1) = 1, \quad T(s_2|s_0, a_2) = 1.$.

The reward combines sparse extrinsic and intrinsic novelty-based components:

$$r(s, a, s') = r^{\text{ext}}(s, a, s') + \beta \cdot r^{\text{int}}(s'),$$

with $r^{\text{int}}(s') = 1/N(s')$, where $N(s')$ is the number of visits to state $s'$.

We assume training with Soft Actor-Critic (SAC), using entropy coefficient $\alpha$. The initial Q-values are uniform, $Q^0(s_i, a_j) = \epsilon$, and the initial policy is uniform: $\pi^0(a_j|s_i) = 0.5$. We consider single-step episodes.

**Definitions and Policy Structure.** The soft Q-function is updated according to the soft Bellman equation:

$$Q^{t+1}(s, a) = r^{\text{ext}}(s, a, s') + \beta \cdot r^{\text{int}}(s') + \gamma V(s'),$$

where the soft value function $V(s)$ is defined as:

$$V(s) = \sum_a \pi(a|s) \left[ Q(s, a) - \alpha \log \pi(a|s) \right].$$

The policy follows a softmax distribution over Q-values:

$$\pi^t(a|s) = \frac{\exp(Q^t(s, a)/\alpha)}{\sum_{a'} \exp(Q^t(s, a')/\alpha)}.$$

**Interaction of Exploration Methods.** Initially, at $s_0$, $Q^0(s_0, a_1) = Q^0(s_0, a_2) = \epsilon$, so the initial policy is:

$$\pi^0(a_j|s_0) = 0.5.$$

Suppose the agent takes $a_1$ and transitions to $s_1$. Then the Q-value becomes:

$$Q^1(s_0, a_1) = \beta \cdot r^{\text{int}}(s_1) + \gamma V(s_1) = \beta + \gamma(\epsilon + \alpha \log 2),$$

assuming $r^{\text{ext}}(s_0, a_1, s_1) = 0$ and $r^{\text{int}}(s_1) = 1/N(s_1) = 1$.

The updated policy becomes:

$$\pi^1(a_1|s_0) = \frac{\exp(Q^1(s_0, a_1)/\alpha)}{\sum_{a'} \exp(Q^1(s_0, a')/\alpha)}, \quad \pi^1(a_2|s_0) = 1 - \pi^1(a_1|s_0).$$

**Analytical Derivation of Step Count $k$.** Define the action probability ratio:

$$\eta^k = \frac{\pi^k(a_1|s_0)}{\pi^k(a_2|s_0)} = \exp\left( \frac{Q^k(s_0, a_1) - Q^k(s_0, a_2)}{\alpha} \right).$$

With repeated visits to $s_1$, the intrinsic reward decays as $r^{\text{int}}(s_1) = 1/k$, leading to:

$$Q^k(s_0, a_1) = \frac{\beta}{k} + \gamma(\epsilon + \alpha \log 2), \quad Q^k(s_0, a_2) = \epsilon.$$

Solving for the step $k^*$ where $\eta^k = 1$ (i.e., equal action probability), we obtain:

$$k^* = \frac{\beta}{(1 - \gamma)\epsilon - \gamma\alpha \log 2}.$$

**Interpretation.** This derived expression for $k^*$ indicates the number of times the agent must revisit state $s_1$ (via action $a_1$) before the action probability ratio $\eta^k$ returns to equilibrium, i.e., $\pi(a_1|s_0) = \pi(a_2|s_0) = 0.5$. At that point, the agent has a higher probability of sampling the action $a_2$ to explore the unvisited state $s_2$.

Initially, high intrinsic rewards bias the agent toward revisiting novel states, resulting in repeated transitions. As novelty decays over time, the effect of entropy increases, driving the policy back toward uniformity. This delayed shift from novelty-based to entropy-driven exploration introduces inefficiencies and slows the agent's ability to discover unvisited regions.

Despite the simplicity of this example, the underlying dynamic—where intrinsic rewards dominate early behavior and decay slowly—applies broadly, including in longer-horizon tasks with more complex novelty mechanisms.

## B. Training Details

We summarize the hyperparameters used in training our method in Tables 5 and Tables 6. Table 5 outlines the default hyperparameters used for both the standard agent $\mathcal{A}^S$ and the novelty-augmented agent $\mathcal{A}^N$ in KEA. These values are kept consistent across tasks unless otherwise noted. Both agents are trained with the Adam optimizer, and architectures and learning rates are adapted for DeepSea.

*Table 5.* **Hyperparameters for KEA.**

| Hyperparameter | $\mathcal{A}^S$ | $\mathcal{A}^N$ |
|---|---|---|
| Actor learning rate | 0.0003 | 0.0003 |
| Critic learning rate | 0.001 / 0.0003 (DeepSea) | 0.001 / 0.0003 (DeepSea) |
| Optimizer | Adam | Adam |
| Actor Architecture | FC(256, 256), FC(64, 64) (DeepSea) | FC(256, 256), FC(64, 64) (DeepSea) |
| Critic Architecture | FC(256, 256), FC(64, 64) (DeepSea) | FC(256, 256), FC(64, 64) (DeepSea) |
| Target Update Frequency | 1 | 1 |
| Target smoothing coeff. $\tau$ | 0.005 | 0.005 |
| Entropy coeff. $\alpha$ | 0.3, 0.1 (DeepSea) | 0.3, 0.1 (DeepSea) |
| Discount factor $\gamma$ | 0.99 | 0.99 |
| Replay buffer size | 300K, 500K (DM Control Suite) 100K (DeepSea) | 300K, 500K (DM Control Suite) 100K (DeepSea) |
| UTD ratio | 1 (2D Navigation Task), 1/2 | 1 (2D Navigation Task), 1/2 |

*Table 6.* **Hyperparameters Across Different Tasks**

| Hyperparameter | 2D Navigation Task | DM Control Suite | DeepSea |
|---|---|---|---|
| Intrinsic architecture (RND) | FC(16, 32) | FC(32, 64) | FC(16, 32) |
| Intrinsic embedding dim. $\hat{f}(s_t; \theta)$ | 16 | 64 | 16 |
| Intrinsic reward clip range | 2 | 2 | 2 |
| Intrinsic reward scale | 0.5 | 0.5 | 0.3 |
| Intrinsic reward Normalization | run mean std | run mean std | run mean std |
| Episode length | 100 | 1,000 | map size (10, 14, 20, 24, 30) |
| Extrinsic reward scale | 100 | 100 | map size (10, 14, 20, 24, 30) |
| Total Environment steps | 300K | 500K | 100K $\times$ map size |
| # Samples before learning | 1,024 | 4,096 | 200 $\times$ map size |
| Batch size | 64 | 64 | 64 |
| Learning rate (RND) | 0.0003 | 0.0003 | 0.0003 |
| Clip gradient norm (RND) | 0.5 | 0.5 | 0.5 |
| Switch Threshold $\sigma$ | 1 | 0.75 | 1 |

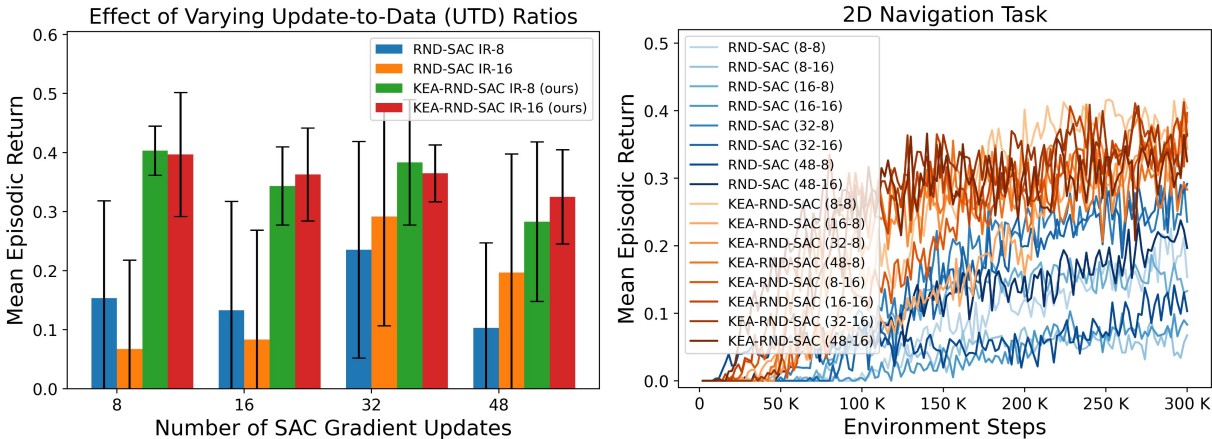

*Figure 7.* Performance under varying update-to-data (UTD) ratios. We vary SAC gradient updates at **8, 16, 32, and 48** times per 32 transitions, and RND updates at **8 and 16** times. *Left*: Final performance, where IR-8 and IR-16 denote RND update at 8 and 16 times, respectively. *Right*: Training curves for all UTD combinations, denoted as **SAC's-RND's** (e.g., 8-16 means the SAC updates 8 times and RND 16 times per 32 transitions).

## C. Analysis of Varying UTD Ratios

The influence of interaction between exploration strategies is not only affected by the exploration strategies mechanism but also influenced by how aggressively the SAC agent and intrinsic reward model are updated. The Update-to-Data (UTD) ratio affects the evolution of both entropy and intrinsic rewards, thereby impacting the shifting between these exploration strategies.

To evaluate KEA's ability to coordinate different exploration strategies and mitigate the inefficiency caused by the complexity of interaction between exploration strategies, we conducted an experiment with varying Update-to-Data (UTD) ratios in the 2D Navigation task (shown in Fig. 3). We compare **KEA-RND-SAC** with **RND-SAC** to evaluate how different UTD ratios (for SAC and RND) affect the overall performance. This comparison highlights how KEA maintains efficient exploration and robustness across a range of UTD ratio settings. We further visualize the training process using a specific example to illustrate how the balance between exploration strategies shifts over time and how these shifts impact exploration performance. Our method demonstrates proactive coordination of exploration strategies, reducing inefficiencies by combining SAC with novelty-based methods, ensuring more consistent and efficient exploration.

### C.1. Experimental Restuls

In this experiment, we varied the UTD ratios by adjusting the number of SAC gradient updates to **8, 16, 32, and 48 times** per 32 transitions, while the number of RND updates was set to either **8 or 16 times**. The goal is to observe how the mean episodic return evolves during training, with a total of 300,000 samples collected from the environment.

As shown in the Fig. 7, KEA-RND-SAC consistently achieves higher mean episodic returns across all UTD ratios when compared to RND-SAC. Specifically, KEA-RND-SAC attains its best performance at $0.403 \pm 0.042$, whereas RND-SAC reaches a lower episodic return of $0.292 \pm 0.197$. However, at the highest UTD ratio (**48 times updates**), both methods experience a performance decline. Despite this drop, KEA-RND-SAC maintains a better performance advantage over RND-SAC. Moreover, KEA-RND-SAC exhibits smaller standard deviations across all configurations, indicating that it is more robust and stable even as the update intensity increases.

### C.2. Visualization

In Fig. 8, we visualize intrinsic rewards, entropy, and action probabilities throughout the training process to illustrate how exploration evolves over time. While RND-SAC successfully reaches the goal in its best cases for both 48 and 8 gradient updates ((a2) and (b2)), it becomes stuck in local minima in the worst cases ((a1) and (b1)), limiting further exploration. In contrast, KEA-RND-SAC consistently reaches the goal across all setups. Compared to RND-SAC, KEA-RND-SAC maintains higher entropy in regions with high intrinsic rewards, especially before reaching the goal. This demonstrates that our method proactively coordinates different exploration strategies (novelty-based exploration via $\mathcal{A}^N$ and stochastic policy

via $\mathcal{A}^S$), thereby reducing the negative effect from the complexity of their interaction. As a result, KEA-RND-SAC ensures more thorough exploration, decreasing the likelihood of getting stuck in local minima.

## D. Details on 2D Navigation Task

The 2D Navigation task is designed to evaluate an agent's ability to explore and reach a sparse-reward goal in a constrained environment. The environment is implemented using the Gymnasium framework (Towers et al., 2024) and consists of a discrete 41×41 grid.

A fixed obstacle of size 4×34 is placed centrally in the grid, effectively creating a narrow passage that the agent must learn to navigate around or through. The goal position is fixed at coordinates (10, 0), located on the right side of the grid. At the beginning of each episode, the agent's starting position is randomly initialized within the left half of the grid.

The observation space is the agent's current $(x, y)$ location in the grid. The action space consists of four discrete actions: move right, move left, move up, and move down. The transition function is unknown to the agent, requiring it to learn effective navigation strategies through trial and error.

Episodes terminate when one of the following conditions is met: the agent reaches the goal, hits the boundary of the environment, or collides with the obstacle. Besides, we truncate the episode when the agent reaches the maximum episode length of 100 steps. The environment provides sparse extrinsic rewards, with a non-zero reward given only when the agent successfully reaches the goal. No intermediate rewards are provided, making exploration especially challenging.

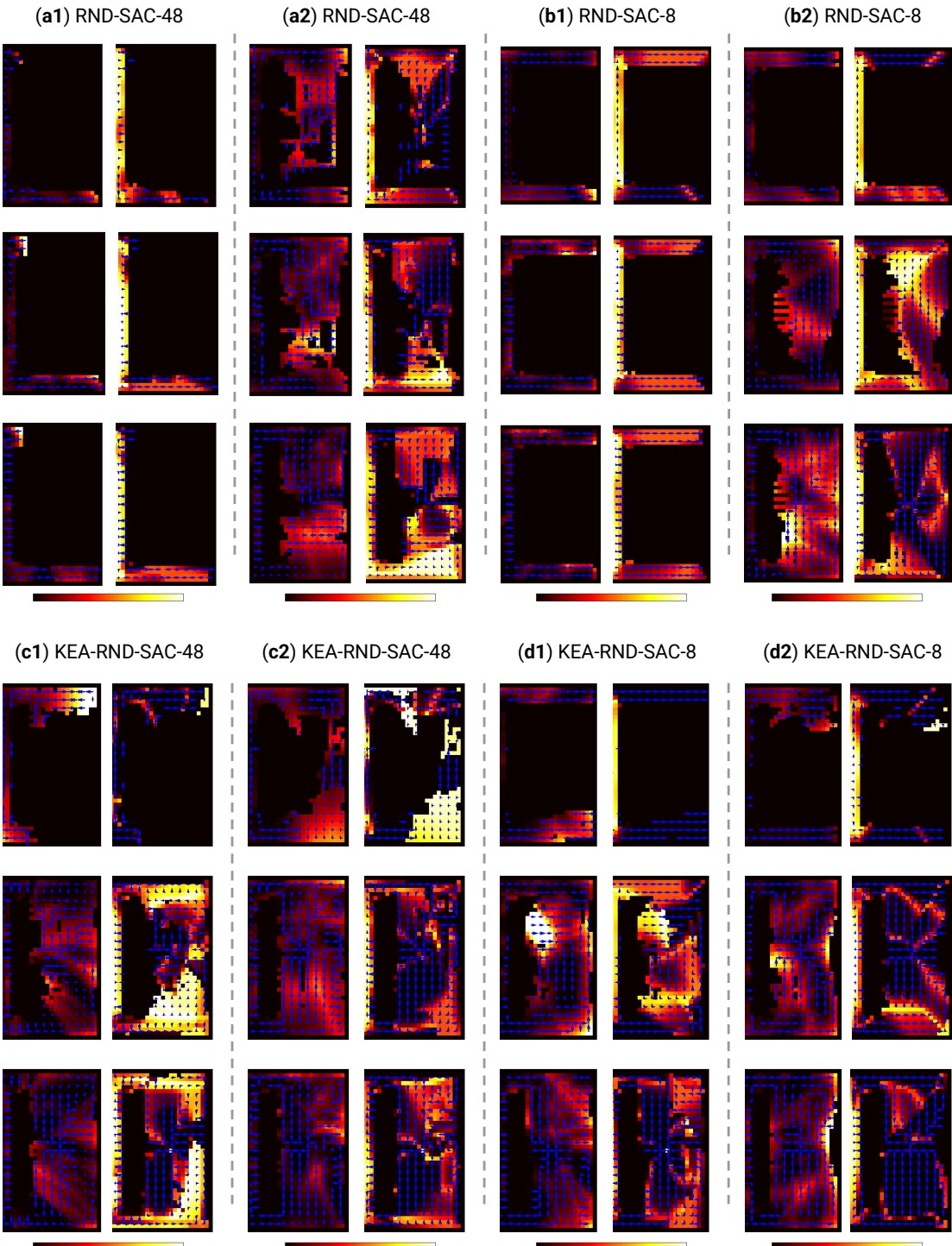

*Figure 8.* Panels (a) and (b) depict RND-SAC using 48 and 8 gradient updates, respectively, while panels (c) and (d) show KEA-RND-SAC under the same conditions. Additionally, (1) highlights the worst performance across five random seeds and (2) highlights the best. In each sub-figure (e.g., (a1)), intrinsic rewards (left) and entropy (right) are presented at three different stages of training: after collecting 20,000, 100,000, and 300,000 samples. Action probabilities are represented by arrows pointing in different directions. For clarity, we focus on the right part of the environment, which showcases the most interesting exploration behaviors, with unexplored states removed. The central obstacle in the environment is shown in Fig. 3.

