# OpenReview forum: "KEA: Keeping Exploration Alive by Proactively Coordinating Exploration Strategies"
_ICML.cc/2025/Conference — ICML 2025 poster_

### Official Review · Reviewer_Numr · 2025-03-11

**Overall Recommendation:** 2

**Summary:**

This paper proposes KEA, an exploration strategy for off-policy RL algorithms such as SAC, DQN, and SQL. Since these algorithms have built-in exploration strategies that interact with novelty-based methods like RND and NovelD, the authors introduce a switching mechanism to decouple them. This mechanism allows the agent to alternate between different exploration strategies based on the state. Experiments on 2D navigation tasks and three tasks from the DeepMind Control Suite demonstrate the superiority of KEA over baseline methods.

**update after rebuttal**

I appreciate the additional experimental results. Due to the lack of some tasks and baselines, as well as the efficiency concerns raised by the need to recompute intrinsic rewards upon each sampling, I would prefer to retain my current score.

**Claims And Evidence:**

From line 45: However, while unvisited states may offer high intrinsic rewards, the agent can not identify them due to a lack of prior experience ... ...

I didn’t fully understand this part. Why is the agent unable to identify unvisited states? According to RND, if certain states have not been visited, the prediction error should be large, making them distinguishable. Could you clarify this point?

**Essential References Not Discussed:**

No

**Experimental Designs Or Analyses:**

Yes. I checked all experimental designs and analyses.

（1）KEA alternates between $A^{B}$ and $A^{SAC}$. Since $A^{SAC}$ is not directly learned by the original SAC algorithm but instead relies on the shaped reward $r_{int} + r_{ext}$, I wonder whether KEA's advantage primarily stems from the shaped rewards. However, I did not find any ablation studies addressing this point. I also noticed experimental results for RND-DQN and RND-SQL, which appear to be based on shaped rewards. If that is the case, why are there no corresponding results for RND-SAC?

（2）For Walker Run Sparse and Cheetah Run Sparse, the thresholds are set at 0.3 and 0.35, respectively. Were these thresholds defined by the original tasks or set by the authors? If they were chosen by the authors, is there a specific rationale or any cited references supporting these choices? Additionally, please clarify that these thresholds were not determined based on experimental results.

(3) Experiments were conducted on only four tasks. The DeepMind Control Suite offers a variety of tasks—why were only three of them selected? Is there a specific reason for this choice?

Additionally, there are other exploration methods beyond RND and NovelD, such as RIDE. While RIDE is mentioned in the Related Work section, it was not included in the experimental comparisons.

**Methods And Evaluation Criteria:**

Yes

**Other Comments Or Suggestions:**

The paragraph starting from line 168 is not very clear. For example, "$A^{B}$ includes a stochastic policy that maintains high variance by slowing down its gradient updates until extrinsic rewards are obtained ...."

The statement seems to imply some details about how to train $A^{B}$. However, since there is no appendix providing the training details, this part of the text causes some confusion.

**Other Strengths And Weaknesses:**

The authors aimed to decouple different exploration strategies, which is a somewhat novel idea that could inspire further research in this direction.

**Questions For Authors:**

A concern with this approach is that existing methods typically use on-policy RL algorithms for RND, partly because shaped rewards may become outdated as the agent explores the environment more thoroughly.

For instance, when the agent first visits a state $S$, its intrinsic reward may be high. However, upon revisiting $S$, the intrinsic reward decreases. In on-policy RL algorithms, these outdated rewards are discarded, whereas in off-policy RL algorithms, they can be reused multiple times. How does KEA ensure that this does not negatively impact the performance?

**Relation To Broader Scientific Literature:**

Exploration is a key challenge in RL research. However, KEA primarily builds on existing methods like RND and NovelD. Moreover, its alternating mechanism simply relies on a threshold that requires tuning. I did not observe any additional novel components in the method.

**Theoretical Claims:**

This paper does not include formal proofs or theoretical claims.

---

> ### Author Rebuttal · Authors · 2025-04-01
>
> We are grateful to Reviewer Numr for the valuable feedback and insightful questions.
>
>
> - **Theoretical Explanation**
>
> We have provided the theoretical explanation in our response to **Reviewer hH71**. Please refer to that section for a detailed discussion.
>
>
> - **Why the Agent Cannot Identify Unvisited States**
>
> Thank you for the insightful question. You are correct that unvisited states yield high intrinsic rewards under RND. However, since SAC updates from transitions stored in the replay buffer, unvisited states--by definition--do not appear in it. As a result, their high intrinsic rewards cannot influence the Q-values or policy updates until the agent actually reaches them. We will revise the text to make this point clearer.
>
>
> - **Comparison with RND-SAC**
>
> Thank you for the question. We did include RND-SAC results—please refer to Figure 3 and Table 1. The baseline labeled "RND" in our experiments is exactly SAC with RND (i.e., RND-SAC). As shown in Section 3.1, KEA (KEA-RND) improves the learning efficiency over RND-SAC. We will revise the notation in the final version to improve clarity and avoid confusion.
>
>
> - **Task-Specific Threshold Selection**
>
> The thresholds for Walker Run Sparse (0.3) and Cheetah Run Sparse (0.35) were set by us, not defined in the original tasks. We chose them to ensure the tasks are sufficiently challenging to highlight differences in exploration ability, without making them so difficult that all methods fail or require excessive training time. These thresholds were not tuned based on experimental results.
>
>
> - **Justification for Excluding RIDE**
>
> Thank you for the suggestion. We chose NovelD as a baseline because it outperforms RIDE in the original paper, and our focus was on evaluating coordination strategies rather than comparing a wide range of exploration methods. Due to space and time constraints, we were unable to include RIDE, but we believe our comparison with NovelD sufficiently demonstrates the effectiveness of KEA.
>
>
> - **Clarification on Supplementary Material**
>
> Thank you for the comment. We do include supplementary material, which provides an analysis of varying UTD ratios and a visualization of intrinsic rewards, entropy, and action probabilities over time to illustrate how exploration behavior evolves during training.
>
> - **Novelty and Contribution Beyond Existing Methods**
>
> Thank you for the feedback. KEA indeed builds on existing exploration methods like RND and NovelD, but our contribution lies in identifying and addressing the coordination issue that arises when combining these with off-policy algorithms like SAC. KEA introduces a lightweight and general switching mechanism that improves exploration efficiency without modifying intrinsic rewards or learning objectives. While threshold-based, the mechanism is simple, effective, and easy to integrate with existing methods. We believe this addresses a practical gap that has been largely overlooked.
>
> - **Clarification on Implementation Details**
>
> We have provided the implementation details in our response to **Reviewer nUkT**. In the final version, we will include additional implementation details in the Appendix to ensure clarity and reproducibility.
>
> - **Update Mechanism for $\mathcal{A}^B$**
>
> We have provided details on the update mechanism for $\mathcal{A}^B$ in our response to **Reviewer nUkT**. Please refer to that section for a complete explanation.
>
>
> - **Handling Outdated Intrinsic Rewards**
>
> Thank you for the insightful question. To address the issue of outdated intrinsic rewards in off-policy settings, we recompute the intrinsic reward for each sampled transition during training, rather than using stored values. This keeps the novelty estimates up to date and mitigates potential negative effects on performance.

---

> > ### Comment · Reviewer_Numr · 2025-04-05
> >
> > While some of my concerns have been addressed, I would still appreciate clarification on why only three tasks from the DeepMind Control Suite were chosen, given the broader set available.

---

> > > ### Author Response · Authors · 2025-04-08
> > >
> > > Thank you for your follow-up. We selected these three tasks from the DeepMind Control Suite as they are commonly used in the literature, facilitating meaningful comparisons. Due to the scope of our study and limited computational resources, we prioritized tasks that balance between complexity and training efficiency. Tasks that are too easy often fail to highlight differences between our method and baselines, while those that are too difficult typically require prohibitively long training times.
> > >
> > > In this work, we evaluate our method on tasks: Walker Run Sparse, Cheetah Run Sparse, and Reacher Hard Sparse. In the rebuttal, we also ran an additional task—Cartpole Swingup Sparse. The result is similar to Reacher Hard Sparse: the episode return curves for KEA-RND and RND are comparable, as both methods are able to obtain extrinsic rewards within a few episodes.
> > >
> > > To further evaluate our approach under hard exploration tasks, in the rebuttal, we also include experiments on DeepSea—a hard exploration benchmark—which complements our DeepMind Control Suite results. The results show that our method achieves comparable performance to SOFE on the easier levels of the DeepSea environments and outperforms SOFE as the difficulty increases (more details in **Reviewer CNJg**).

---

### Official Review · Reviewer_CNJg · 2025-03-13

**Overall Recommendation:** 1

**Summary:**

This paper proposes KEA, which aims to balance novelty-based exploration and SAC’s inherently stochastic-based exploration. The authors argue that naively combining novelty-based exploration with SAC results in suboptimal performance. To address this issue, KEA introduces a co-behavior agent that works alongside SAC and a switching mechanism to facilitate proactive coordination between exploration strategies from the two methods. The proposed approach is evaluated on a 2D navigation task and a subset of sparse reward control tasks from the DeepMind Control Suite.

## Update after rebuttal
I appreciate additional experimental results and explanations. However, my concerns regarding the experimental results remain. The performance of the proposed approach is still mixed; in many tasks, the baselines achieve similar or better results. Furthermore, the claim that the proposed approach could "operate on the between explored and unexplored area" is not sufficiently supported by the evidence provided.

**Claims And Evidence:**

The reviewer found that several claims lack sufficient support from the experimental results. For instance, the claim on lines 180-181, stating that the coordination ensures consistent escape from local minima, is not substantiated by any quantitative or qualitative evidence demonstrating such an escape. Similarly, the claim on lines 203-205, suggesting that the mechanism ensures operation near the boundary between explored and unexplored regions, lacks evidence to confirm this capability. While these may be the intended goals of the proposed approach, the reviewer notes that the evidence provided does not convincingly demonstrate their achievement. The reviewer suggests that the authors provide additional evidence or revise their claims to better align with the experimental results.

More importantly, the reviewer has some concerns about the experimental settings. Please see the ‘Experimental Designs Or Analyses’ sections for more details.

**Essential References Not Discussed:**

None

**Experimental Designs Or Analyses:**

The experiments section lacks enough supporting evidence. Please see details below.

1. The reviewer finds that the experimental section of the paper provides limited evidence and is not convincing. Specifically, the reviewer notes that only four tasks (one 2D maze and three tasks from DeepMind Suites) were used to evaluate the proposed approach. This limited number of tasks makes it difficult to properly assess the effectiveness and generalizability of the method.  To address this limitation, the reviewer suggests that the authors include results on a more comprehensive set of tasks, such as additional tasks from the DeepMind Control Suite, to provide a more thorough and convincing evaluation of the proposed approach.

2. In the limited reported experimental results, the performance of the proposed approach appears mixed. Specifically, as shown in  Table 1 and Table 3, the baseline Novel1D seems to outperform the proposed approach on 2D Navigation and reacher hard sparse. The mixed results raise concerns about the effectiveness of the proposed method.

3. The baseline approach used in the experiments, while classic, appears to be somewhat outdated (RND 2018 and NovelID 2021). Comparison with state-of-the-art methods, such as [a] and [b], is essential to demonstrate the effectiveness and advancements of the proposed approach.

[a] Rethinking Exploration in Reinforcement Learning with Effective Metric-Based Exploration Bonus, Wang et al, 2024.

[b] Improving Intrinsic Exploration by Creating Stationary Objectives, Castanyer et al, 2024

**Methods And Evaluation Criteria:**

The primary technical innovation presented in this work is the switching mechanism, which alternates between the two exploration policies based on the intrinsic reward received. However, the reviewer  finds this technical contribution to be somewhat limited in its scope and significance.


For Evaluation Criteria, please see the ‘Experimental Designs Or Analyses’ sections.

**Other Comments Or Suggestions:**

Typos: L152 (left) As figure 2, …

**Other Strengths And Weaknesses:**

The reviewer finds that the clarity of the paper needs improvement, as many technical details require further clarification. For example, Figure 1 is confusing: it lacks an explanation of what the colors in the map and the gray bar in the middle of the map represent, and the distinctions between each figure are unclear. These elements need to be adequately addressed to enhance the overall clarity and comprehensibility of the paper.

The statement "slowing down its gradient updates until extrinsic rewards are obtained" (L173-175) lacks clarity. The authors need to provide a formal statement and explain how this mechanism functions.

**Questions For Authors:**

Please see the above discussions.

**Relation To Broader Scientific Literature:**

This paper aims to unify entropy-based exploration, as used in SAC, with intrinsic reward-driven methods, such as RND.

**Theoretical Claims:**

No formal theoretical claim is presented.

---

> ### Author Rebuttal · Authors · 2025-04-01
>
> We greatly appreciate Reviewer CNJg for the helpful comments and thoughtful suggestions.
>
>
> - **Theoretical Explanation**
>
> We have provided the theoretical explanation in our response to **Reviewer hH71**. Please refer to that section for a detailed discussion.
>
> - **Broader Contribution Beyond the Switching Mechanism**
>
> Thank you for the feedback. Our contribution goes beyond proposing a switching mechanism--we identify and analyze a core inefficiency in combining SAC with novelty-based exploration, and design KEA as a practical coordination solution. The method improves exploration consistency, is lightweight, and integrates easily with existing approaches.
>
> - **Faster Convergence as a Key Strength of KEA**
>
> Thank you for the comment. While final returns in some tasks (e.g., 2D Navigation, Reacher Hard Sparse) are similar between KEA-NovelD and NovelD, KEA-NovelD consistently achieves faster convergence. For instance, in 2D Navigation, KEA-NovelD reaches a return of 0.6 around 190k steps, whereas NovelD requires 250k steps, as shown in the experimental results. This demonstrates KEA’s advantage in sample efficiency, even when final performance is close.
>
>
> - **Comparison with Recent State-of-the-Art Methods in DeepSea**
>
> Thank you for the helpful suggestion. We agree that comparing against recent state-of-the-art exploration methods is important to demonstrate the effectiveness of KEA. To address this, we conducted additional experiments in the **DeepSea** environment [1], evaluating our method (KEA-RND-SAC) against the recently proposed SOFE (Castanyer et al., 2024), as well as DeRL (Schäfer et al., 2021).
>
> DeepSea is a hard-exploration benchmark defined on an $N \times N$ grid, where only penalized rightward actions lead to the goal—posing a significant credit assignment challenge for agents relying solely on extrinsic rewards.
>
> Following the setup in SOFE, the table below summarizes average returns and one standard deviations over 100,000 evaluation episodes:
>
> | Algorithm   | DeepSea 10       | DeepSea 14       | DeepSea 20       | DeepSea 24       | DeepSea 30       |
> |------------|------------------|------------------|------------------|------------------|------------------|
> | DeRL-A2C   | **0.98 ± 0.10**  | 0.65 ± 0.23      | 0.42 ± 0.16      | 0.07 ± 0.10      | 0.09 ± 0.08      |
> | DeRL-PPO   | 0.61 ± 0.20      | 0.92 ± 0.18      | -0.01 ± 0.01     | 0.63 ± 0.27      | -0.01 ± 0.01     |
> | DeRL-DQN   | **0.98 ± 0.09**  | **0.95 ± 0.17**  | 0.40 ± 0.08      | 0.53 ± 0.27      | 0.10 ± 0.10      |
> | SOFE-A2C   | 0.94 ± 0.19      | 0.45 ± 0.31      | 0.11 ± 0.25      | 0.08 ± 0.14      | 0.04 ± 0.09      |
> | SOFE-PPO   | 0.77 ± 0.29      | 0.67 ± 0.33      | 0.13 ± 0.09      | 0.07 ± 0.15      | 0.09 ± 0.23      |
> | SOFE-DQN   | 0.97 ± 0.29      | 0.78 ± 0.21      | 0.70 ± 0.28      | **0.65 ± 0.26**  | **0.42 ± 0.33**  |
> | **KEA-RND-SAC**| 0.97 ± 0.05      | 0.89 ± 0.06      | **0.73 ± 0.13**  | **0.66 ± 0.12**  | **0.43 ± 0.31**  |
>
> These results show that KEA-RND-SAC achieves comparable or superior performance to SOFE across varying levels of difficulty in the DeepSea environment. We will include these new results in the final version to strengthen the empirical evaluation and highlight KEA’s effectiveness in addressing complex exploration challenges.
>
> -----
> [1] Behaviour suite for reinforcement learning, Osband et al., 2019
>
> [2] Improving Intrinsic Exploration by Creating Stationary Objectives, Castanyer et al, 2024
>
> [3] Decoupled Reinforcement Learning to Stabilise Intrinsically-Motivated Exploration, Schäfer, Lukas, et al., 2021
>
>
> - **Improving the Clarity of Figure 1**
>
> Thank you for the valuable feedback. We agree that additional explanation would improve the clarity of Figure 1. In the final version, we will clarify the meaning of the color map, the gray bar (which represents an obstacle), and the differences between each subfigure. Each subfigure shows the agent’s behavior in a specific region at different time stages, illustrating how intrinsic rewards and policy entropy shape exploration. We will revise the figure and caption to make this clearer.
>
> - **Clarification on Update Mechanism for $\mathcal{A}^B$**
>
> Thank you for the comment. During training, we set the loss weight for $\mathcal{A}^B$ to zero until extrinsic rewards are obtained, effectively freezing its updates and maintaining high action variance. Once extrinsic rewards are observed, we set the loss weight to one to gradually train $\mathcal{A}^B$. This allows $\mathcal{A}^B$ to remain exploratory early on and focus on task optimization later. We agree that providing a formal statement and clearer explanation will improve clarity, and we will revise this part accordingly in the final version.
>
> - **Correction of Typographical Error**
>
> Thank you. We will correct the sentence in L152.

---

### Official Review · Reviewer_hH71 · 2025-03-22

**Overall Recommendation:** 2

**Summary:**

This paper presents KEA, a RL-based to enhance exploration efficiency in sparse reward environments. The authors propose a proactive coordination mechanism between novelty-based exploration methods and the stochastic policy of Soft Actor-Critic . KEA introduces a co-behavior agent and a dynamic switching mechanism to maintain exploration diversity, improve learning efficiency, and mitigate redundant sample collection.

**Claims And Evidence:**

Evidence includes experimental results from 2D navigation tasks and continuous control tasks from the DeepMind Control Suite, demonstrating improved performance and faster convergence compared to baselines.

**Essential References Not Discussed:**

Some bandit-based exploration strategies are missing. For example, (1) Neural contextual bandits with ucb-based exploration; (2) Ee-net: Exploitation-exploration neural networks in contextual bandits; (3) Neural thompson sampling.

**Experimental Designs Or Analyses:**

Experiments include a 2D navigation task and three continuous control tasks from the DeepMind Control Suite.

**Methods And Evaluation Criteria:**

The core method includes an additional co-behavior agent, operating alongside SAC, with a dynamic switching mechanism for exploration based on state novelty measured by intrinsic rewards.

**Other Comments Or Suggestions:**

none

**Other Strengths And Weaknesses:**

Strengths: (1) Clear methodological contributions with empirical validation; (2) Effective visualizations illustrating key concepts.

Weaknesses: (2) The novelty may not surpass the bar of ICML; adding theoretical analysis is helpful to improve this paper.   (2) Primarily limited to off-policy RL settings; limited applicability for on-policy methods.

**Questions For Authors:**

How sensitive is KEA to different novelty computation methods beyond RND and NovelD?

Can the propose exploration method process a decent theoretical performance guarantee?

**Relation To Broader Scientific Literature:**

Builds upon existing novelty and curiosity-based exploration methods.

**Theoretical Claims:**

No theoretical claims.

---

> ### Author Rebuttal · Authors · 2025-04-01
>
> We are grateful to Reviewer hH71 for the constructive and valuable feedback.
>
> - **Theoretical Explanation of Exploration Strategy Interaction Problem**
>
> > **Problem Setup and Assumptions**
> >
> > Consider an MDP with states $S=$ { $s_0, s_1, s_2$ }, actions $A=$ { $a_1, a_2$ }, and deterministic transitions: $T(s_1|s_0, a_1)=1$, $T(s_2|s_0, a_2)=1$.. Rewards combine sparse extrinsic and intrinsic novelty-based components:
> $$
> r(s, a, s') = r^{ext}(s, a, s') + \beta \ r^{int}(s')\ ,
> $$
> >
> > with $r^{int}(s')=1/N(s')$, where $N(s')$ counts state visits. We assume Soft Actor-Critic (SAC) with entropy coefficient $\alpha$, uniform initial Q-values ($Q^0(s_i,a_j)=\epsilon$), uniform initial policy ($\pi^0(a_j|s_i)=0.5$), and single-step episodes.
>
> > **Definitions and Policy Structure**
> >
> > The soft Q-function, denoted as $Q(s, a)$, is updated according to the following soft Bellman operator, which incorporates both intrinsic and extrinsic rewards:
> $$
> Q^{t+1}(s, a) = r^{ext}(s, a, s') + \beta \ r^{int}(s') + \gamma\ V(s')
> $$
> >
> > where the soft value function $V(s)$ is given by:
> $$
> V(s) = \sum_a \pi(a|s)\ [Q(s, a) - \alpha\ \log \pi(a|s)]
> $$
> >
> > The policy probabilities are given by a softmax of the Q-values:
> $$
> \pi^{t}(a|s) = \frac{exp(Q^{t}(s, a)/ \alpha)}{\sum_{a'} exp(Q^{t}(s, a')/ \alpha)}
> $$
>
>
> > **Interaction of Exploration Methods**
> >
> > Initially at state $s_0$, $Q^0(s_0,a_1)=Q^0(s_0,a_2)=\epsilon$, thus $\pi^0(a_j|s_0)=0.5$. After taking action $a_1$ and transitioning deterministically to $s_1$, the updated Q-value becomes:
> $$
> Q^{1}(s_0, a_1)
> = \beta \ r^{int}(s_1) + \gamma\ V(s_1)
> = \beta + \gamma\ (\epsilon + \alpha \log2)
> $$
> >
> > assuming $r^{ext}(s_0,a_1,s_1)=0$, and initial intrinsic reward $r^{int}(s_1)=1 / N(s_1)=1$.
> >
> > The updated policy probabilities at $s_0$ are then:
> $$
> \pi^{1}(a_1|s_0) = \frac{exp(Q^{1}(s_0, a_1)/ \alpha)}{\sum_{a'} exp(Q^{1}(s_0, a')/ \alpha)}\ ,\ \
> \pi^{1}(a_2|s_0) = 1 - \pi^{1}(a_1|s_0)
> $$
>
> > **Analytical Derivation of Step Count $k$**
> >
> > Define the action probability ratio at step  as:
> $$
> \eta^k = \frac{\pi^{k}(a_1|s_0)}{\pi^{k}(a_2|s_0)}
> = exp(\frac{Q^{k}(s_0, a_1) - Q^{k}(s_0, a_2)}{\alpha})
> $$
> >
> > With repeated visits to state , intrinsic rewards reduces as $r^{int}(s_1) = 1/k$, giving:
> $$
> Q^{k}(s_0, a_1)
> = \frac{\beta}{k}+\gamma\ (\epsilon+\alpha \log(2)), \ \ Q^{k}(s_0, a_2) = \epsilon
> $$
> >
> > For equal probabilities ($\eta^k = 1$), solving explicitly yields:
> $$
> k^* = \frac{\beta}{(1-\gamma)\epsilon-\gamma \alpha \log(2)}
> $$
>
> > **Interpretation**
> >
> > The derived equation explicitly demonstrates how the intrinsic reward factor ($\beta$) influences equilibrium behavior and highlights constraints on valid parameter ranges.
> >
> > Initially, intrinsic rewards increase the probability of revisiting novel states, resulting in repeated collection of similar transitions. This effect reduces as the state novelty decays, and action probabilities return to equilibrium (0.5). The delayed shift from novelty-driven to entropy-driven exploration may thus introduce inefficiencies and slow down learning. Despite the simplified setup, this dynamic is likely broadly relevant, including cases with longer episodes and more complex novelty-based intrinsic rewards.
>
>
> - **Explanation of How KEA Addresses the Interaction Problem**
>
> As discussed in our theoretical explanation, combining SAC with novelty-based exploration can lead to inefficiencies—such as repeatedly visiting a state (e.g., $s_1$) to reduce its novelty and lower the action probability ratio $\eta$. KEA addresses this by switching to the co-behavior agent, whose $\eta$ is already close to 1, enabling more efficient coordination without excessive additional sampling.
>
> - **On-policy Limitation**
>
> Thank you for the suggestion. We focused KEA on off-policy RL due to challenges in combining novelty-based exploration with off-policy methods, especially in transfer across continuous control tasks. KEA addresses these issues by coordinating exploration strategies, improving efficiency and performance. Adapting KEA to on-policy settings would require reconsidering or redesigning the interaction between $\mathcal{A}^{SAC}$ and $\mathcal{A}^{B}$, especially since a shared replay buffer is no longer available in such frameworks.
>
> - **Comparison with Recent State-of-the-Art Methods in DeepSea**
>
> To further strengthen our empirical evaluation, we performed additional experiments in the DeepSea environment—an established benchmark for hard-exploration tasks. The results show that KEA-RND-SAC performs on par with or better than recent state-of-the-art methods. For more details, please refer to our response to **Reviewer CNJg**.

---

### Official Review · Reviewer_Acaj · 2025-03-23

**Overall Recommendation:** 3

**Summary:**

This paper proposes an exploration technique for sparse reward problem. They propose to use a co-behavior agent to reduce the interference when combining two exploration mechanism. In particular, they have the co-behavior agent to perform novelty-based exploration while the standard agent explores through traditional stochastic policy. Further, they introduce a switching mechanism to dynamically select between the two agent.

**Claims And Evidence:**

They claim that novelty based exploration technique may interfere with the implicit exploration mechanism of stochastic policy. While they provide an intuitive explanation through the heat-map of action probabilities, this lacks a theoretical grounding. Experimental results resonate with the claims.

**Essential References Not Discussed:**

The paper should discuss a bit on ensemble-based exploration methods that often use multiple agents.

**Experimental Designs Or Analyses:**

Overall, the experiments are well designed. In Figure 5, SAC results are missing in first two experiments. Also, further experiments on different navigation tasks or high-fidelity environments that require exploration would benefit the paper.

**Methods And Evaluation Criteria:**

Assuming the discussed interference, the proposed approach appears as an interesting and simple solution. However, there is no evaluation/comment on the overhead (time and memory) introduced by the additional agent.

**Other Comments Or Suggestions:**

I found it pretty confusing why the authors name the novelty-based agent as A^SAC and the traditional stochastic agent as A^B. I would suggest to alter the notation, denoting the entropy-based traditional agent as A^SAC and naming other agent accordingly.

In line 264, "This demonstrates that our method not only maintains exploration efficiency but also improves convergence speed". I believe instead it is better to attribute improved exploration for faster convergence.

**Other Strengths And Weaknesses:**

Experimental results with other off-policy methods clearly indicates the wide applicability of this method.

The writing could be improved especially in terms of clarity.

**Questions For Authors:**

Since there is a trade-off in selecting \sigma, is there any findings or prescription to select \sigma under different reward structure? Or it needs to be manually tuned as the problem (reward structure) changes.

**Relation To Broader Scientific Literature:**

I found this work interesting and orthogonal to the existing works. Disentangling different exploration mechanism with separated network (agent) nicely juxtaposes with the current literature that mostly incorporates novelty/count based exploration mechanism within the same network.

**Theoretical Claims:**

The paper doesn't provide any theoretical insights. It would be nice to see a theoretical backup of why natural shifts between novelty-based exploration and SAC’s stochastic policy-based exploration may result in delays and inefficiencies.

---

> ### Author Rebuttal · Authors · 2025-04-01
>
> We thank Reviewer Acaj for the valuable suggestions and helpful comments.
>
>
> - **Theoretical Explanation**
>
> We have provided the theoretical explanation in our response to **Reviewer hH71**. Please refer to that section for a detailed discussion.
>
>
> - **Computational Overhead of the Additional Agent**
>
> Both agents share a unified replay buffer, which improves data efficiency and limits memory overhead. Training is slightly slower due to computing losses for both $\mathcal{A}^{SAC}$ and $\mathcal{A}^B$. At inference time, only one agent ($\mathcal{A}^B$) is active, and the action computation is equivalent to a standard SAC policy, so runtime and memory overhead remain minimal.
>
>
> - **Comparison with Recent State-of-the-Art Methods in DeepSea**
>
> Thank you for the suggestion. We conducted additional experiments in the DeepSea environment, a well-established hard-exploration benchmark. In these experiments, KEA-RND-SAC was compared against recent state-of-the-art methods and demonstrated comparable or superior performance. For more details, please refer to our response to **Reviewer CNJg**.
>
>
> - **Clarification on Missing SAC Results in Figure 5**
>
> Thank you for the feedback. SAC results in Figure 5 are always remain zero. Without novelty-based exploration, SAC fails in these hard exploration tasks.
>
>
> - **Clarification on Notation of Agents**
>
> Thank you for the suggestion. We agree that the current naming could be confusing and will revise the notation in the final version to improve clarity and readability.
>
> - **Clarification on the Relationship Between Exploration and Convergence**
>
> Thank you for the suggestion. We agree that the faster convergence observed in our method is a result of improved exploration. We will revise the sentence accordingly in the final version to better reflect this relationship.
>
>
> - **Limitations and Future Directions in Threshold Selection**
>
> Lower thresholds encourage more stochastic exploration, while higher ones rely more on novelty-based exploration. The threshold balances these two strategies and is task-dependent. In our experiments, we hand-tuned the threshold to keep the usage of $\mathcal{A}^{SAC}$ around 85–90%. While automated methods (e.g., grid search, Bayesian optimization) could optimize this threshold, defining a general and adaptive selection mechanism remains a non-trivial challenge and a promising direction for future work.

---

### Official Review · Reviewer_nUkT · 2025-03-23

**Overall Recommendation:** 3

**Summary:**

This paper introduces KEA (Keeping Exploration Alive), a method designed to address coordination issues that arise when combining Soft Actor-Critic (SAC) with novelty-based exploration methods. The research identifies that when SAC's stochastic policy exploration coexists with novelty-based exploration, the complex interactions between these strategies can lead to exploration inefficiencies and redundant sampling.

The paper's main contribution is proposing a mechanism to proactively coordinate different exploration strategies: (1) introducing a co-behavior agent ($\mathcal{A}^\text{B}$) that works alongside SAC incorporating existing novelty-based methods ($\mathcal{A}^\text{SAC}$); and (2) designing a dynamic switching mechanism ($ψ$) based on state novelty that intelligently determines which strategy to use. When novelty exceeds a threshold, the system switches to the co-behavior agent to maintain high-entropy exploration; when novelty falls below the threshold, the system employs the SAC agent for exploring relatively novel regions.

## update after rebuttal

After carefully reviewing the authors' responses, I find that they have adequately addressed my main concerns. I have decided to update my rating to weak accept.

**Claims And Evidence:**

Several core claims in the paper lack sufficient supporting evidence:
1. Exploration Strategy Interaction Problem: Although the paper identifies the issue of inefficiency caused by interaction between exploration strategies through Figure 1 and 2D navigation experiments, it lacks theoretical proof.
2. Effectiveness of KEA: The paper claims KEA effectively coordinates exploration strategies, but testing on only 3 DeepMind Control environments is insufficient to support claims of broad effectiveness.
3. Generalizability of KEA: The claim that KEA can be generalized to other off-policy methods is supported by limited evidence from simple tests with DQN and SQL. Particularly for cases where the effect is not clear (e.g., standard DQN), the paper fails to provide theoretical explanations.
4. Design Rationale for Coordination Mechanism: The paper proposes a threshold-based switching mechanism but lacks theoretical analysis or extensive experiments to support this design over other possible coordination approaches. The basis for threshold selection is also insufficiently justified.

While the paper presents interesting problems and solution approaches, it lacks comprehensive and in-depth experimental evidence and theoretical support, with many key claims requiring more substantial evidence for verification.

**Essential References Not Discussed:**

The paper fails to cite CAT-SAC [1], which addresses nearly identical problems with a complementary approach. While KEA introduces a co-behavior agent with a switching mechanism, CAT-SAC directly modifies SAC's entropy temperature to be curiosity-aware, making it higher in unfamiliar states and lower in familiar ones.

References:

[1] Lin, Junfan, et al. "Cat-sac: Soft actor-critic with curiosity-aware entropy temperature." (2020).

**Experimental Designs Or Analyses:**

The authors conducted experiments in a 2D navigation task and three sparse reward environments from the DeepMind Control Suite, but the experimental design has significant limitations. First, the 2D navigation task, which serves as the primary concept validation platform, is severely under-described, lacking critical details about environment parameters, reward design, and state space. Although Figure 8 provides some visualization results, these are difficult to comprehensively evaluate and interpret due to insufficient background information about the task.

In the DeepMind Control Suite experiments, the authors selected three environments: Walker Run Sparse, Cheetah Run Sparse, and Reacher Hard Sparse. Results show that KEA-RND and KEA-NovelD indeed outperform their unmodified counterpart algorithms, providing some support for the method's effectiveness. However, selecting only three highly homogeneous locomotion control tasks as evaluation benchmarks is clearly insufficient and fails to demonstrate the method's effectiveness across more diverse environments. Particularly questionable is the lack of exploration into the method's performance in non-sparse reward environments, leaving unanswered whether KEA would introduce unnecessary computational overhead or even negative effects in standard reward settings.

Regarding the threshold selection experiments, the authors tested different thresholds (0.50 to 1.50) and their impact on KEA's performance. Results show that threshold settings affect the frequency of co-behavior agent usage, with higher thresholds leading to lower usage rates, while a threshold of σ=1.00 achieved optimal performance. The paper lacks theoretical analysis explaining why specific thresholds perform better and doesn't provide a general methodology for threshold selection. This lack of theoretical guidance for parameter selection means that threshold tuning in practical applications may require extensive experimentation, increasing the difficulty of applying the method.

The UTD ratio experiment is one of the more complete analyses in the paper, where the authors explored algorithm performance under different update intensities by adjusting the update frequency of SAC and RND (8 to 48 times). KEA-RND can maintain higher policy entropy in regions with high intrinsic rewards, reducing the likelihood of getting stuck in local optima. Figure 8 visualizes the evolution of exploration behavior under different UTD settings, intuitively illustrating how KEA coordinates different exploration strategies.

**Methods And Evaluation Criteria:**

The method proposed in the paper conceptually aligns with the problem it aims to solve (the coordination issue when combining SAC with novelty-based exploration methods), but there are significant deficiencies in the evaluation criteria: (1) The 2D navigation task experimental environment lacks critical details, making it impossible to assess or reproduce. (2) The evaluation is limited to only 3 sparse reward environments from DeepMind Control Suite, which is insufficient to demonstrate the method's broad applicability. (3) Comparisons are restricted to original SAC, SAC-RND, and SAC-NovelD, without benchmarking against a wider range of state-of-the-art exploration methods.

**Other Comments Or Suggestions:**

I recommend the following improvements to strengthen the paper:
1. Expand the background section to include a more thorough explanation of maximum entropy RL, which would help readers better understand the problem the paper aims to solve.
2. Provide detailed configuration information for both $\mathcal{A}^\text{B}$ and $\mathcal{A}^\text{SAC}$, including network architecture, parameter sizes, learning rates, and discount factors to ensure reproducibility.
3. Include comprehensive details about the 2D navigation task setup, including state/action spaces, reward function design, and environmental constraints.
4. Add pseudocode for the proposed algorithm to clarify the implementation details and make the method more accessible to other researchers.

**Other Strengths And Weaknesses:**

Weaknesses:
1. Limited Experimental Evaluation: The paper only evaluates the method on three DeepMind Control sparse reward environments and one 2D navigation task, which is insufficient to fully demonstrate the method's generalizability across different problem domains.
2. Lack of Theoretical Foundation: The paper provides no theoretical justification for why the proposed switching mechanism works effectively.
3. Insufficient Comparison Baselines: The comparison with state-of-the-art methods is limited, missing opportunities to benchmark against other exploration coordination approaches.
4. Inadequate Environment Details: The 2D navigation task, which serves as a motivating example, lacks comprehensive description of its reward function, state representation, and action space, making it difficult to assess the method's effectiveness or reproduce the results.
5. Threshold Selection: The method relies on manually tuned threshold $σ$ without providing an automatic selection mechanism, potentially limiting its applicability to new environments.
6. Missing Implementation Details: The paper omits critical details about the configuration of $\mathcal{A}^\text{B}$ and $\mathcal{A}^\text{SAC}$, such as network architecture, parameter sizes, learning rates, and discount factors, which are essential for reproduction.

Strengths:
1. Elegant Solution Approach: The paper proposes a conceptually simple yet effective method for coordinating different exploration strategies, addressing a practical problem in reinforcement learning with sparse rewards.
2. Insightful Visualizations: The visualizations effectively demonstrate how the proposed method maintains exploration in high-novelty regions compared to baseline approaches.

**Questions For Authors:**

1. Could you elaborate on the update mechanism for the policy $\mathcal{A}^\text{B}$? The paper mentions it follows the standard SAC update, but more details would be helpful.
2. Please provide more information about the "includes a stochastic policy that maintains high variance by slowing down its gradient updates until extrinsic rewards are obtained" mechanism for $\mathcal{A}^\text{B}$. What specific approach is used to slow down these updates, and how were the parameters for this mechanism determined?

**Relation To Broader Scientific Literature:**

KEA builds upon the long-standing challenge of sparse rewards in reinforcement learning. In recent years, intrinsic reward methods have become mainstream solutions, including curiosity-driven [1] and novelty-driven [2,3] approaches. KEA does not propose an entirely new intrinsic reward mechanism, but rather identifies and addresses a specific problem that arises when combining these methods with maximum entropy reinforcement learning algorithms.

A core insight of the paper relates to the interaction between entropy-regularized algorithms like SAC [4] and novelty-based exploration. SAC optimizes both exploration and exploitation by maximizing policy entropy in the objective function, an idea that originates from early maximum entropy reinforcement learning work [5]. This paper points out that in regions with high intrinsic rewards, policies tend to become deterministic, conflicting with the entropy maximization objective and leading to inefficient exploration.

KEA's primary innovation lies in proposing an exploration strategy coordination mechanism, which relates to several research directions:
1. Hierarchical Reinforcement Learning: KEA's co-behavior agent (AB) and switching mechanism resemble high-level policies in hierarchical RL [6], but focus on exploration strategies rather than task decomposition.
1. Multi-Policy Learning: Similar to the Options framework [7], KEA employs multiple policies, but its purpose is to solve a specific exploration strategy interaction problem.

References:

[1] Pathak, Deepak, et al. "Curiosity-driven exploration by self-supervised prediction." International Conference on Machine Learning. PMLR, 2017.

[2] Burda, Yuri, et al. "Exploration by Random Network Distillation." International Conference on Learning Representations, 2019.

[3] Badia, Adrià Puigdomènech, et al. "Never Give Up: Learning Directed Exploration Strategies." International Conference on Learning Representations, 2020.

[4] Haarnoja, Tuomas, et al. "Soft Actor-Critic: Off-Policy Maximum Entropy Deep Reinforcement Learning with a Stochastic Actor." 2018.

[5] Ziebart, Brian D., et al. "Maximum entropy inverse reinforcement learning." AAAI, 2008.

[6] Barto, Andrew G., and Sridhar Mahadevan. "Recent advances in hierarchical reinforcement learning." Discrete event dynamic systems 13 (2003): 341-379.

[7] Sutton, Richard S., Doina Precup, and Satinder Singh. "Between MDPs and semi-MDPs: A framework for temporal abstraction in reinforcement learning." Artificial intelligence 112.1-2 (1999): 181-211.

**Theoretical Claims:**

The coordination mechanism design lacks theoretical foundation: the switching mechanism appears to be based on intuition rather than mathematical analysis, and lacks theoretical justification for why this design is superior to other possible approaches.

---

> ### Author Rebuttal · Authors · 2025-04-01
>
> We sincerely thank Reviewer nUkT for the thoughtful and detail feedback.
>
> - **Theoretical Explanation**
>
> We have provided the theoretical explanation in our response to **Reviewer hH71**. Please refer to that section for a detailed discussion.
>
> - **Rationale Behind the Threshold-Based Switching Mechanism**
>
> We provide a detailed theoretical explanation in our response to **Reviewer hH71**. Instead of requiring repeated visits to reduce the action probability ratio $\eta$ (e.g., visiting $s_1$ multiple times), KEA switches to the co-behavior agent, which already maintains a high $\eta$ value. This allows for more efficient and timely coordination between exploration strategies without the need for additional sampling.
>
>
> - **Comparison with Recent State-of-the-Art Methods in DeepSea**
>
> Thank you for the thoughtful feedback. We agree that broader evaluation is important to support the effectiveness and generality of KEA. To address this, we conducted additional experiments in the DeepSea environment, a widely used benchmark for hard-exploration tasks. These experiments evaluate KEA-RND-SAC against recent state-of-the-art methods, including SOFE (Castanyer et al., 2024) and DeRL (Schäfer et al., 2021). The results show that KEA achieves comparable or superior performance across increasing levels of task difficulty, reinforcing the value of its coordination mechanism in challenging exploration settings. For detailed results and discussion, please refer to our response to **Reviewer CNJg**.
>
>
>
> - **Why KEA Is Not Effective with DQN**
>
> In DQN, $\epsilon$-greedy exploration is independent of Q-values, so the issue KEA addresses—delayed shifting from novelty-driven to method's original exploration—does not arise. Therefore, KEA is not expected to provide benefits in this case. We included DQN mainly to highlight this contrast and emphasize that KEA is more suited to methods, where exploration is influenced by intrinsic reward-driven Q-values.
>
> - **Clarification on the 2D Navigation Task Setup**
>
> Thank you for the feedback. The task description is provided in Section 3.1. To clarify further: the environment is a 41×41 grid with a 4×34 obstacle in the center; the goal is fixed at (10, 0), and the agent starts randomly in the left half. The maximum episode length is 100 steps and the episode terminated when reach the goal, the boundary, or the obstacle. A reward is given only when reaching the goal. We will improve clarity and reproducibility in the final version.
>
>
> - **Limitations and Future Directions in Threshold Selection**
>
> Lower thresholds encourage more stochastic exploration, while higher ones rely more on novelty-based exploration. The threshold balances these two strategies and is task-dependent. In our experiments, we hand-tuned the threshold to keep the usage of $\mathcal{A}^{SAC}$ around 85–90%. While automated methods (e.g., grid search, Bayesian optimization) could optimize this threshold, defining a general and adaptive selection mechanism remains a non-trivial challenge and a promising direction for future work.
>
>
> - **Relation to CAT-SAC and Distinction from KEA**
>
> Thank you for pointing out CAT-SAC, which we will cite. While both methods aim to improve coordination in novelty-driven SAC, CAT-SAC modulates entropy temperature based on novelty, whereas KEA introduces a co-behavior agent and a switching mechanisum. A key advantage of KEA is that the co-behavior agent is trained without intrinsic rewards and thus reliably converges to the optimal policy for the extrinsic task.
>
> - **Clarification on Implementation Details**
>
> Thank you for the feedback. Both $A^B$ and $\mathcal{A}^{SAC}$ use Fully connected(256, 256) actor and Q-network. Learning rates are 0.0003 for Actors and 0.001 for Q-networks, with a discount factor of 0.99. We will include full configuration details for $A^B$, $\mathcal{A}^{SAC}$, and novelty-based models in the Appendix in the final version.
>
> - **Update Mechanism for $\mathcal{A}^B$**
>
> Thank you for the question. $\mathcal{A}^B$ is updated jointly with $\mathcal{A}^{SAC}$. In each update step, we sample a batch from the replay buffer and compute the soft Q-value losses and policy losses for both $\mathcal{A}^{SAC}$ and $\mathcal{A}^B$ separately, then sum them for optimization. Notably, before any extrinsic reward is received (i.e., before task completion), the loss weight for $\mathcal{A}^B$ is set to zero.
>
> - **Mechanism for Maintaining High Variance in $\mathcal{A}^B$**
>
> During training, we set the loss weight for $\mathcal{A}^B$ to zero until extrinsic rewards are obtained, effectively freezing its updates and maintaining high action variance. Once extrinsic rewards are observed, we set the loss weight to one to gradually train $\mathcal{A}^B$. This allows $\mathcal{A}^B$ to remain exploratory early on and focus on task optimization later.
>
>
> - **Including Pseudocode**
>
> Thank you for the feedback! We will add pseudocode for KEA in the final version.

---

> > ### Comment · Reviewer_nUkT · 2025-04-05
> >
> > I appreciate the authors' detailed responses to my concerns. While KEA demonstrates promising results on exploration-focused tasks, I remain uncertain about its effectiveness across a broader range of continuous action space environments. I would like to see evaluation on more DeepMind Control Suite tasks to verify whether KEA performs equal to or better than standard SAC in environments that don't specifically focus on exploration challenges.
> >
> > Additionally, the manual threshold tuning remains a significant limitation. Even with grid search or Bayesian optimization, determining appropriate thresholds for new environments would require substantial computational resources, limiting the method's practical applicability.
> >
> > Based on these remaining concerns, I maintain my current rating (leaning towards reject).

---

> > > ### Author Response · Authors · 2025-04-08
> > >
> > > Thank you for your thoughtful feedback and for engaging deeply with our work. We would like to clarify a few key points regarding the scope and design choices behind KEA.
> > >
> > > Our primary motivation is to improve performance on hard exploration tasks, where standard RL algorithms like SAC typically struggle. Novelty-based exploration is commonly used in such cases to enhance exploration. KEA is specifically designed to address challenges in hard exploration tasks, by combining novelty-based exploration with SAC in a better way. Therefore, while we understand the interest in broader evaluations, environments that do not present significant exploration challenges are not the main focus of our study.
> > >
> > > To further evaluate our approach under hard exploration tasks, in the rebuttal, we also include experiments on DeepSea—a hard exploration benchmark—which complements our DeepMind Control Suite results. The results show that our method achieves comparable performance to SOFE on the easier levels of the DeepSea environments and outperforms SOFE as the difficulty increases (more details in **Reviewer CNJg**).
> > >
> > > Regarding the need to set the switching threshold $\sigma$, we found that setting $\sigma$ around 1 is generally effective and robust across different tasks. This follows common practice in novelty-based exploration methods, where intrinsic rewards are normalized using a running mean and standard deviation, leading to a distribution with mean near 0 and standard deviation close to 1. As a result, $\sigma = 1$ serves as a reasonable default, without requiring extensive tuning.
> > > Moreover, as shown in Table 2, a small discrete set of values, such as ${0.5, 0.75, 1, 1.25, 1.5}$, is sufficient for tuning when necessary, with a step size of 0.25. In our experiments, for each new environment, we run a single episode and select the $\sigma$ value that leads to an approximately 15% usage rate of the co-behavior agent ($A^B$), which serves as a practical and efficient estimate. This procedure helps limit the computational cost.
> > >
> > > We appreciate your constructive comments and hope this helps clarify our design choices and the intended scope of our contributions.

---

### Decision · Program_Chairs · 2025-05-01

**Decision:**

Accept (poster)

**Comment:**

This paper presents KEA (Keeping Exploration Alive), a method that addresses exploration inefficiencies when combining Soft Actor-Critic (SAC) with novelty-based exploration methods. The authors identify that interactions between SAC's stochastic policy and novelty-based exploration can lead to delayed shifts between exploration strategies and redundant sample collection. KEA introduces a co-behavior agent alongside SAC and a switching mechanism to coordinate exploration strategies based on state novelty.

After carefully reviewing the submission, the reviews, and the subsequent discussions, I recommend a **weak acceptance** for this paper. The submission identifies an issue in reinforcement learning exploration, specifically how the interaction between SAC's inherent stochastic policy and novelty-based exploration methods can lead to inefficiencies. The authors propose a conceptually simple yet effective coordination mechanism to address this problem. The reviewers agreed that the paper addresses a practical problem in reinforcement learning exploration. Reviewer nUkT noted that KEA does not propose an entirely new intrinsic reward mechanism but rather identifies and addresses a specific coordination problem between exploration strategies. Similarly, Reviewer Acaj found the work "interesting and orthogonal to the existing works," appreciating how it disentangles different exploration mechanisms.

A common concern among reviewers was the limited theoretical foundation in the initial submission. Both Reviewers hH71 and nUkT requested theoretical justification for the switching mechanism design. The authors partially addressed this in their rebuttal by providing a more formal analysis of the exploration strategy interaction problem, demonstrating how intrinsic rewards can influence policy probabilities and lead to inefficient exploration without proper coordination. The experimental evaluation generated mixed opinions. While the paper demonstrated improved learning efficiency in sparse reward settings, Reviewers CNJg and Numr questioned the limited scope of experiments (three DeepMind Control tasks plus a 2D navigation task). In their rebuttal, the authors provided additional results on the DeepSea environment, comparing KEA with more recent state-of-the-art methods like SOFE. These additional experiments somewhat strengthened the empirical evaluation, though Reviewer CNJg remained unconvinced about some of the paper's stronger claims regarding exploration coordination.

The manual threshold tuning requirement was identified as a practical limitation by multiple reviewers. Reviewer nUkT specifically called this "a significant limitation" for the method's practical applicability. The authors clarified in their response that the threshold typically requires minimal tuning, with σ=1.0 being a reasonable default since intrinsic rewards are typically normalized using running statistics. While this explanation provides some practical guidance, the concern about the need for manual parameter selection remains valid.